# Multicomponent reactions provide key molecules for secret communication

Andreas C. Boukis[1], Kevin Reiter [2], Maximiliane Frölich[1], Dennis Hofheinz[3] & Michael A.R. Meier [1]

A convenient and inherently more secure communication channel for encoding messages via specifically designed molecular keys is introduced by combining advanced encryption standard cryptography with molecular steganography. The necessary molecular keys require large structural diversity, thus suggesting the application of multicomponent reactions. Herein, the Ugi four-component reaction of perfluorinated acids is utilized to establish an exemplary database consisting of 130 commercially available components. Considering all permutations, this combinatorial approach can unambiguously provide 500,000 molecular keys in only one synthetic procedure per key. The molecular keys are transferred nondigitally and concealed by either adsorption onto paper, coffee, tea or sugar as well as by dissolution in a perfume or in blood. Re-isolation and purification from these disguises is simplified by the perfluorinated sidechains of the molecular keys. High resolution tandem mass spectrometry can unequivocally determine the molecular structure and thus the identity of the key for a subsequent decryption of an encoded message.

[1] Laboratory of Applied Chemistry, Institute of Organic Chemistry (IOC), Karlsruhe Institute of Technology (KIT), Straße am Forum 7, Karlsruhe 76131, Germany. [2] Institute of Nano Technology (INT), Karlsruhe Institute of Technology (KIT), Hermann-von-Helmholtz-Platz 1, Eggenstein-Leopoldshafen 76344, Germany. [3] Institute for Theoretical Informatics (ITI), Karlsruhe Institute of Technology (KIT), Am Fasanengarten 5, Karlsruhe 76131, Germany. Correspondence and requests for materials should be addressed to M.A.R.M. (email: m.a.r.meier@kit.edu)

In the digital age, data security, in all its facets from every day applications via professional environments to espionage, is of utmost importance. Although ultimate security will never be reached, a constant progress and improvement of cryptography and steganography, both evolving with the needs and possibilities of new technologies on the one hand and the steady increase of the capabilities of adversaries on the other hand, is required. A core task of cryptography is encryption, i.e., to convert messages into unintelligible ciphertexts that can only be decrypted by a receiver who possesses a dedicated decryption key. Steganography, in contrast, hides the fact that a message (encrypted or not) is being transmitted. For a higher degree of security, decryption keys can be concealed by steganography, i.e., via chemicals[1–3].

The idea to use chemicals for secret communication dates back to ancient times when secret inks were first applied[4]. Today, more sophisticated methods are discussed for data encryption, such as fluorescent materials[5–11] or multi-analyte fluorescent molecular sensors[1, 12–14]. Furthermore, DNA was exploited for secure communication[15–19]. Molecular logic gates[20–25], molecular computing systems[26–28], and systems based on authorizing password entries[29–38] also contributed to this field. Other approaches utilized NMR chemical shifts[39], microorganism colonies[40], antibodies[41], 3D photonic crystals[42], and molecular tags (equipped with halogen substituted aromatic sidechains) serving as barcodes for the identification of chemical libraries via gas chromatography[43].

Considering cryptography, state-of-the-art symmetric encryption schemes, like the Advanced Encryption Standard (AES, also known as Rijndael)[44], Serpent, or Blowfish can protect data reliably, but require a secret key for their operation. This secret key must be known by both the encryptor and the decryptor, and should be chosen uniformly at random. Typically, secret keys are short (e.g., 128 bits), but it is not a priori clear how encryptor and decryptor receive common secret keys. Popular methods to distribute secret keys involve the use of asymmetric cryptography (e.g., using the Diffie-Hellman key exchange scheme, or a key transport mechanism based on an asymmetric encryption scheme, such as, RSA). The disadvantage of using asymmetric cryptography (besides being considerably less efficient than symmetric cryptography) is that state-of-the-art asymmetric encryption schemes require specific algebraic structures, and thus may potentially be more prone to a structured cryptanalysis.

However, hiding of the secret key via steganography (e.g., via chemicals) can introduce an additional level of security. Even if the number of known chemical compounds is tremendously large and steadily growing, their application as molecular keys is limited considering certain requirements: First, a systematic synthesis methodology should be applied providing molecules with a high level of structural complexity. Moreover, simple, robust, and reproducible procedures should be used. Finally, the products should be thermally and chemically as inert as possible and be designed for simple isolation, purification, and analysis.

Multicomponent reactions (MCR) represent highly suitable synthetic tools to provide molecular keys, fulfilling these criteria. In a MCR, three or more precursor components are combined to one reaction product, containing moieties of all precursors[45, 46]. In the field of MCRs, the Ugi reaction was appealing, because four components are combined to a single product in a straightforward one-pot reaction[47]. In the Ugi reaction (see Fig. 1a and Supplementary Figure 1 for the reaction mechanism), an aldehyde (green) reacts with an amine (orange), a carboxylic acid (blue), and an isocyanide (red) to from a *bis*-amide Ugi product displaying four individually defined sidechains ($R^{1–4}$, introduced by the four starting components). Variation of the precursor components creates molecular diversity with minimal synthetic effort[48–50].

Herein we report a means to advance chemical communication systems (including the inspiring work of Margulies et al.[1]), via a secret communication channel based on molecular keys, which can be easily hidden in various media, transferred nondigitally[1], isolated via F-tags and unambiguously read out via ESI-MS/MS (one of the most developed and sensitive analytical techniques). In our system, the well-established AES algorithm is combined with an effective hiding and transportation of the encryption key. The latter consists of systematically selected and tailor-made molecules that can be transported in a concealed fashion via a non-digital channel.

## Results

**Design and synthesis of molecular keys.** For exploring the synthetic potential of the Ugi reaction, an exemplary list of components (see Supplementary Data 1), including ten perfluorinated carboxylic acids, 50 aldehydes, 50 amines, and 20 isocyanides, was designed. The components chosen for this database are commercially available and selected in order to selectively react to the desired Ugi products (criteria listed in the Methods section and illustrated in Supplementary Figure 1). This set of 130 components can potentially be combined to $10 \times 50 \times 50 \times 20 = 500{,}000$ different molecular keys. The number of components is only limited by the availability of components suitable for the Ugi reaction and can be extended easily beyond this set of commercially available compounds. The main function of the list of components is to assign chemical information (i.e., reacting components and side chains of the respective molecular keys) to alphanumerical codes (i.e., systematic combinations of letters, numbers, and special characters). For this purpose, a letter is given to a certain chemical functional group, and the different sidechains within the same category of functional groups are counted with arbitrary numbers, e.g., aldehydes → letter A, benzaldehyde → A(001), butyraldehyde → A (003), …; isocyanides → letter B, …). The list of components is highly flexible, because the alphanumerical assignment can be exchanged or adjusted if necessary.

As proof of principle, a sub-library of different molecular keys was synthesized by systematic variation of the different reacting components, utilizing the Ugi reaction of perfluorinated acids. Considering molecular design, the molecular keys were equipped with a perfluorinated side chain (also called F-Tag), enabling a highly simplified purification via fluorous solid phase extraction (F-SPE)[51, 52]. F-SPE can retain fluorous molecules selectively and hence separate the molecular keys from organic contaminates and/or matrix materials.

In Fig. 1, the variation of components is illustrated in a 3D plot for one of the applied perfluorinated acids. Each point represents a herein synthesized, distinct molecule. The coordinates are linked to the components used and thus the sidechains of the molecular key. Two performed systematic component variations, unambiguously demonstrating the synthesis possibility with different components, are represented in the expansions (Fig. 1c, d). The molecular keys were analyzed via 1D and 2D nuclear magnetic resonance spectroscopy (NMR, see Supplementary Figure 2), high resolution mass spectrometry (HRMS) and infrared spectroscopy to confirm their structure and thus to demonstrate the viability of the present library design (synthetic procedures and analysis can be found in the Supplementary Methods).

Our molecular keys provide a robust steganographic channel that can be used to transport, i.e., cryptographic keys. The transmitted keys can hardly be recognized, because (i) an adversary does not know that a key is hidden in a molecule; (ii) only the recipient knows where the molecular key is located/stored (i.e., adsorbed on paper, dissolved in perfume, see below); (iii) information on

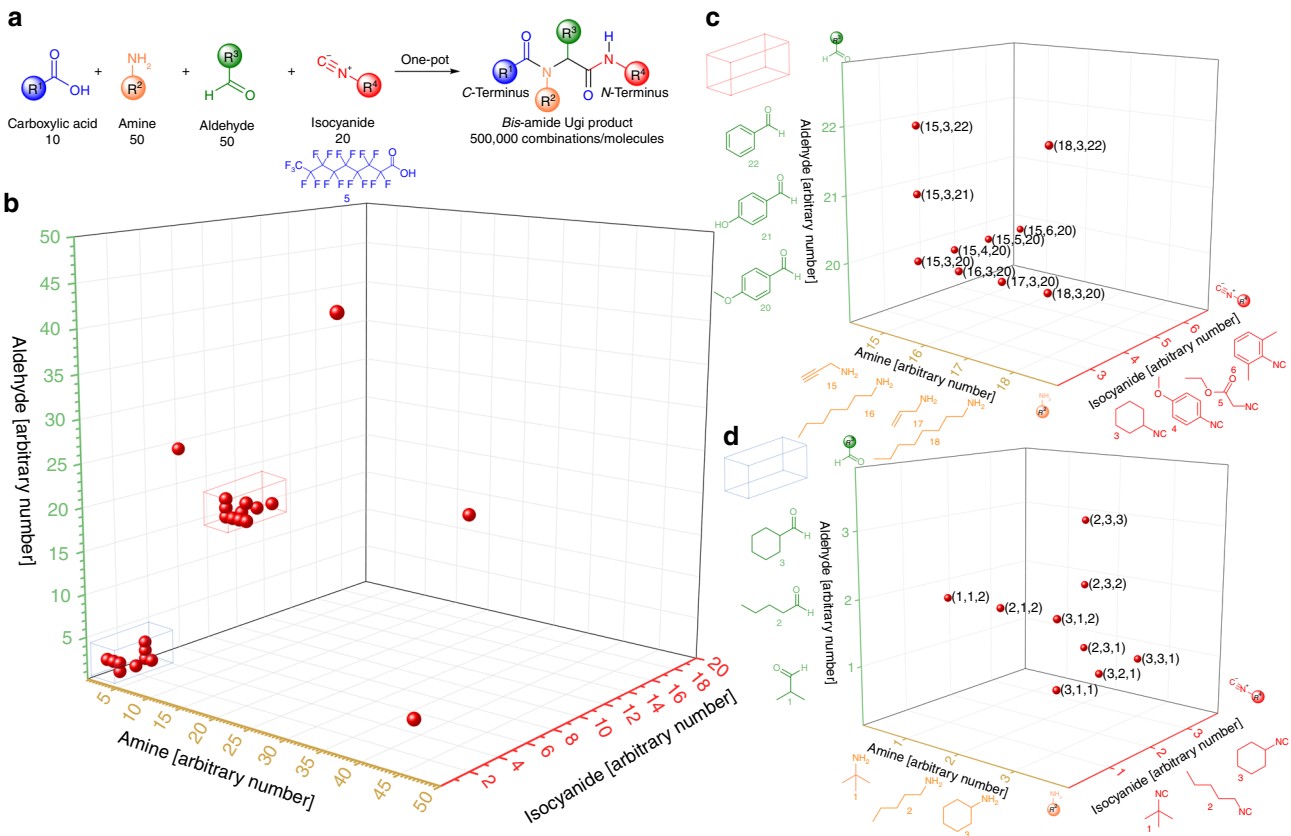

**Fig. 1** Molecular keys. **a** Schematic representation of the Ugi reaction. The combinatorial scope of components considered in the exemplary list of components involves 500,000 molecules. **b** 3D plot of a set of molecules synthesized with perfluorononaic acid (shorter and longer acids were also evaluated). The axes represent different functional groups. The numbers on the axes represent different components (i.e., sidechains). **c**, **d** Expansions of the two systematic variations highlighted by the red and blue box in **b**, respectively

how to extract and analyze the molecule, convert the chemical structure into digital information and apply the decryption is necessary; and (iv) since the keys are transferred physically, an attacker cannot rely on computing power alone.

Data storage capacity is admittedly much higher for DNA than for synthetic molecules, which might change in the near future with the development of sequence defined macromolecules[53–55]. The user can thus choose the appropriate system for the required application. Importantly, the molecular keys are designed for simplified purification utilizing fluorous tags, as well as for straightforward characterization (tandem-MS), requiring only trace amounts of the key at all stages.

An exemplary way of key transmission is presented in Fig. 2. The sender and recipient meet for the first time and exchange details on how to secretly communicate in the future. They need to agree on how the molecular key will be transferred (e.g., adsorbed on paper, dissolved in a perfume, etc.), whether the decryption key will be fragmented onto several molecular keys (and if so how many individual molecular keys will be used) in order to increase/adjust the required level of security. Additionally, the list of components, should be exchanged and the analytical methods are preselected. These initially communicated details should remain secret, however, if an adversary reveals one or some of the discussed details, he is still not able to decrypt the message without knowing the other information and most importantly the molecular keys.

**Coding, transportation, and isolation of molecular key.** The encryption of secret documents/files (examples included in

Supplementary Data 2) is performed by an independent state-of-the-art symmetric encryption algorithm such as AES (herein a random 128-bit AES key). Currently, people, organizations, military, and governments benefit from AES encryption for protecting, e.g., classified information, email communication, opening Virtual Private Network (VPN) tunnels for secure internet connections, online banking, or secure file transfer protocols such as Hypertext Transfer Protocol Secure (HTTPS). However, the molecular key strategy offers a universal steganographic channel, and is not limited to AES applications, therefore opening new perspectives for every symmetric encryption scheme. The ciphertexts are transferred conventionally in a digital channel (e.g., via email or even shared publicly) or in a traditional letter. Decrypting encrypted data without the corresponding key is infeasible with keys of sufficient entropy. The respective encryption keys are concealed in molecular keys and transported non-digitally via a steganographic chemical channel (Fig. 2a) (see below for transfer examples). The cryptographic digital channel can, in principle, transmit data of any size. The steganographic channel for the molecular keys is smaller in terms of data size and utilized for key distribution exclusively. If the herein presented list of components is accessible for the adversary, there are still 500,000 possible combinations (~18 bits). For higher levels of security, the AES encryption key can be fragmented and transferred onto several molecular keys, which are transferred independently (spatially- or time-displaced, e.g., the first molecular key is transferred underneath the stamp, the second one in the top left corner of the letter, and third one is included in a perfume, etc.). Thus, if six molecular keys are utilized, the key size equals ~113 bits (100 bits are considered

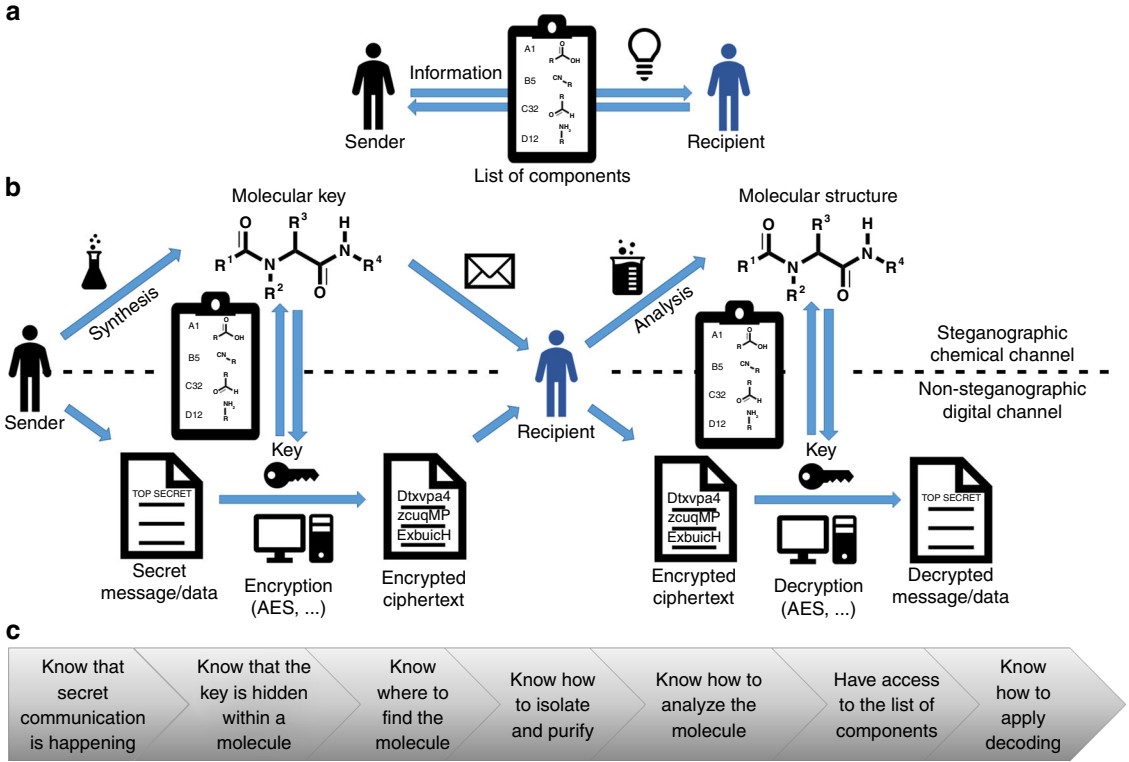

**Fig. 2** Steganographic key distribution. **a** The sender and recipient discuss details about their secret communication and exchange the list of components (assigns chemical information to alphanumerical symbols). **b** from left to right: The sender synthesizes one or several molecular keys and determines the alphanumeric code associated with the chemical structures according to the list of components. The alphanumeric code, e.g., A(001)-B(012)-C(007)-D(007) is serving as encryption key. The molecular key is concealed and transferred to the recipient via a steganographic channel. The recipient isolates the key and elucidates the molecular structure for decryption. **c** Levels of security and knowledge an adversary needs for decrypting the message

sufficiently secure in cryptography). Additionally, such a key fragmentation increases the steganographic security level, since an adversary would have to intermit and decode each individually transferred molecular key.

As a first transportation example, an envelope (representative for adsorption onto paper, see Fig. 3 and Supplementary Figure 3) was selected. A small quantity (4 mg) of the molecular key was dissolved in a minimal amount methanol and transferred onto the top right corner of the envelope with a pipet. After drying, the molecular key was adsorbed onto the large surface of the cellulose fibers, not traceable for the bare eye. For further protection and simple recognition of the area covered with the molecular key, a stamp (preferably self-adhesive) was placed above the treated area.

After the letter reached its destination, the covered area was cut into pieces and extracted with methanol and dichloromethane. Since the obtained material might contain impurities (which could interfere with later analysis), the molecular key was purified via F-SPE, further demonstrating the major advantage of the herein used F-tagged molecular keys. The molecular key was obtained in excellent purity and sufficient quantity for subsequent analysis (see Methods section). Alternatively, the molecular key was concealed and transported in a commercial perfume, adsorbed onto instant coffee powder, green tea, or on sugar (see Methods section for procedures and Supplementary Figure 3 for pictures). To further demonstrate the robustness of our isolation protocol, a molecular key was hidden in and reisolated from a blood sample. If a cryptographic key coded in DNA or another biomacromolecule was to be hidden in blood, the ubiquitously present native DNA, proteins, nutrients, and many other components in this complex mixture can cause severe

complications for the readout. Hence, these successful reisolation examples clearly demonstrate the advantages of the herein presented molecular keys. The presented secret communication channels could be utilized by secret service agencies, for safe data transmissions in companies or the communication between individuals or institutions in a totalitarian state regime, to only name some scenarios that require higher levels of security.

**Reading the molecular key via structure elucidation**. Analytical chemistry offers a variety of methods for the identification of the chemical compounds and thus the readout of molecular keys. The molecular structure is unambiguously solved if the four sidechains ($R^{1-4}$) are determined. In mathematic terms, a four-dimensional problem is solved considering four different parameters. The four parameters are obtained via four different molar masses from high resolution tandem mass spectrometry (the monoisotopic mass + three fragments). Tandem mass spectrometry (tandem-MS) can be applied universally, minimal sample preparation is necessary and the detection limits are very small, thus requiring only trace amounts of substance to synthesize, transport and obtain the information. In Fig. 4, a representative tandem-MS spectrum is presented, wherein the symbols mark different species (fragments or the intact molecule) utilized for evaluation. Fragment assignment determines the sidechains of the molecular key and decrypts the message. In an exemplary scenario, the sender would synthesize one or several molecular keys and determine the alphanumeric code associated with the chemical structures according to the list of components. In a particular case, the components perfluoropentanoic acid, 4-methoxyphenylisocyanide, pentylamine, and benzaldehyde

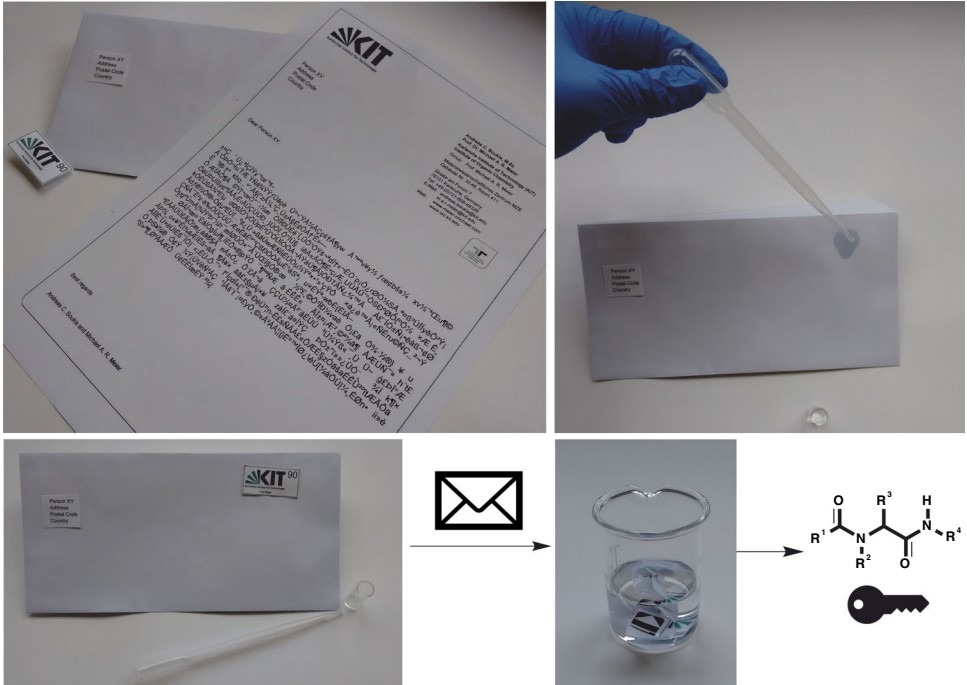

**Fig. 3** Transportation and extraction of the molecular key. The encrypted message is exemplarily transferred in form of a letter. The molecular key is dissolved and placed in the top right corner of the envelope and afterwards covered with a stamp. After the letter has been sent and received, the molecular key is extracted with a solvent and purified by F-SPE to subsequently analyze the molecule and encrypt the message. Additional images are included in Supplementary Figure 3

were chosen and converted into A(001)-B(012)-C(007)-D(007). This alphanumeric code is utilized to encrypt a message or data. The recipient isolates and analyzes a molecular key via ESI-MS and determines a $m/z$ of 595.1615 Da for $[M + Na]^+$. After entering this mass into the analysis script (Supplementary Software 1), the most probable molecules and their respective fragments are listed. Fragmenting the $m/z = 595$ Da species via ESI-MS/MS generates characteristic fragments at $m/z = 176.14$ and 106.06 Da, which are entered into the script and are sufficient information to identify the molecular structure of the molecular key (Fig. 4). Using the list of components, the corresponding alphanumeric code can now be generated and utilized to decrypt the message/data (more examples can be found in Supplementary Data 2). A detailed procedure on how to read out the molecular keys is provided in the Methods section. Supplementary Data 2 contains three examples of encrypted messages and the tandem-MS spectra of the molecular keys for decryption. Upon identification of the prominent fragmentation patterns, the four fragments displayed in Fig. 5a were chosen for structure elucidation. In order to facilitate the readout, an analysis script (Supplementary Software 1) provides the most probable fragments after the exact mass of the molecule is entered. Since not every signal from the tandem-MS spectrum is required for the readout (unlabeled signals in Fig. 4), the analysis script is very useful for the readout.

The monoisotopic mass is a valuable indication for the molecular structure. In Fig. 5b, a database analysis based on the list of components is displayed. The frequency of occurring masses within a $\Delta M$ threshold of 0.01 Da (which is well within the accuracy of modern high-resolution mass spectrometers regarding the investigated molecular weights) are displayed in a distribution plot (more details in Supplementary Figure 4). It can be concluded that preselecting the monoisotopic mass reduces the number of remaining possible molecules drastically (at least by three orders of magnitude). With the fragments obtained

via tandem-MS, the molecular structure is determined and transferred into alphanumerical codes via the list of components. The alphanumerical code is entered into the molecular encryption script (Supplementary Software 2) for decryption of the sent message. As recommended, the final AES code can also be encoded onto several molecular keys, which are transferred independently (separated spatially or time-wise). In the case of these fragmented keys, the individual alphanumerical code of each molecular key is entered in the corresponding sequence into the encryption script.

In conclusion, the herein presented molecular key strategy allows steganographic key distribution in combination with a flexible and adaptable data safety protocol. Molecular keys with a data storage capacity of ~18 bit were synthesized in a one-pot reaction approach. The molecular structures were analyzed by a combination of tandem-MS fragmentation and computer assisted readout (analysis script, Supplementary Software 1). The respective structures served as decryption keys for AES encrypted messages or data files (Supplementary Software 2). In principle, the herein presented keys could also be used for other applications, e.g., as identification or anti counterfeit tags. In future research, the investigation of molecular keys with even higher data storage capacities will be of great interest in order to overcome current limitations of the system.

## Methods

**Criteria for the list of components.** For the list of components presented in Supplementary Data 1, we decided to exclusively focus on commercially available components. In addition, only aldehydes were chosen as carbonyl species because they are more reactive in Ugi reactions than ketones. However, the scope of carbonyl components reported in literature is larger (but if other functional groups are introduced they first should be tested for compatibility). The components need to fulfill certain requirements (graphically illustrated in Supplementary Figure 1c). The components should only carry one functional group that can participate in the Ugi reaction (i.e., a carboxylic and an aldehyde moiety should not be combined in the same precursor molecule, otherwise polymerization might occur). The

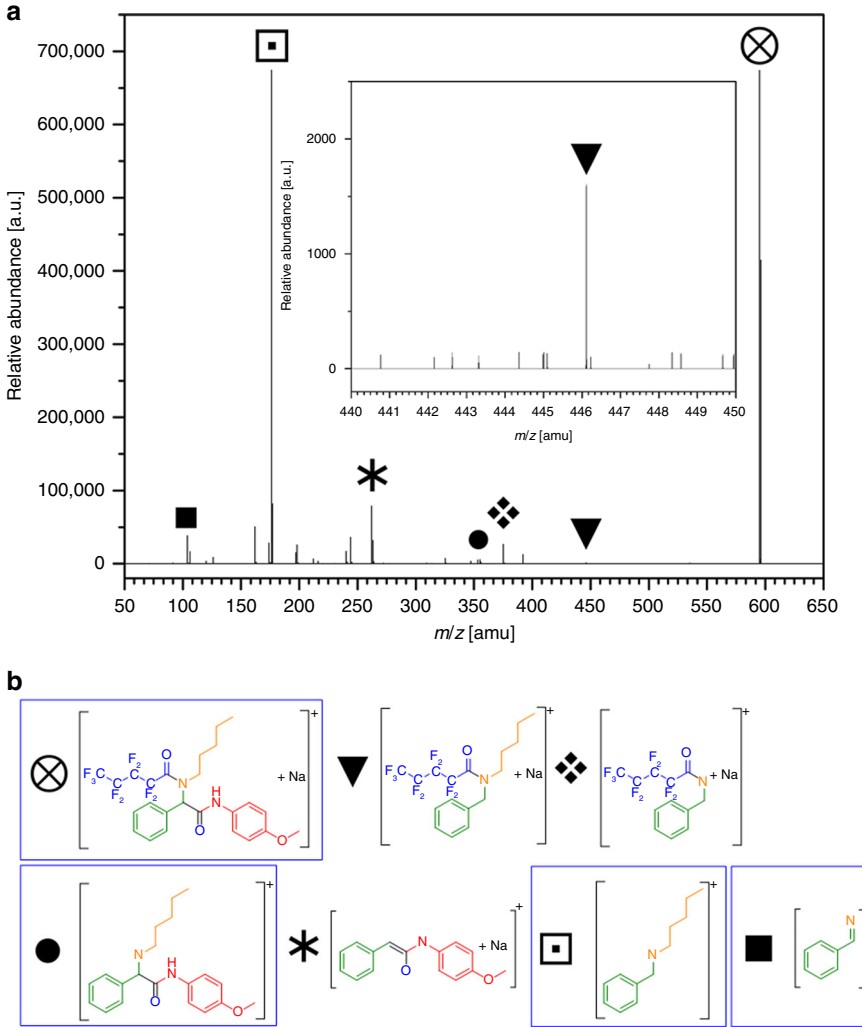

**Fig. 4** Readout of molecular keys via tandem-MS. **a** exemplary ESI-MS/MS spectrum. **b** Proposed fragment assignment. The monoisotopic mass of the intact molecule (circle containing cross) is the first information that significantly reduces the number of possibilities. Additionally, three fragments are necessary to unambiguously determine the combination of components. The three fragments marked in the blue frame (black circle, white square containing black square, and black square) are utilized in the computer analysis script (Supplementary Software 1) for assisted readout. The other fragments (black inverted triangle, diamond cluster, and asterisk) further confirm the structure, but are not essential for a successful readout

components should not carry moieties, which are strongly redox sensible, photo-sensitive, or easily hydrolyzed in the presence of protic solvents like methanol or water. In this context, we additionally excluded *tert*-butyl esters from our database because they are deprotected at low pH by trifluoracetic acid. Functional groups, which are known to cause side reactions with any other substance involved in the Ugi reaction, should also be avoided. In addition, phenolic hydroxyl groups on electron-poor aromatic systems (i.e., carrying additional nitro substituents) should be avoided due to undesired side reactions (i.e., Ugi-Smiles where the acidic phenol acts as acid component). Since the molecular keys are designed for tandem mass spectrometric analysis, the dataset for one component (same functional group) should not contain isomers (e.g., no aldehydes with the same molar mass). Isotope labeled components can be beneficial for MS purposes, but were also excluded from our database. Diastereomeric mixtures of molecular keys synthesized from racemic starting components can be treated and utilized similar to stereomerically pure molecular keys (a study commenting on the influence of stereochemistry is included in Supplementary Methods).

### General synthetic procedures

*General procedure (GP1) for Ugi reactions: larger scale, higher efficiency.* In a 25 mL round bottom flask the aldehyde (1.70 eq., typically 1 mmol) was dissolved in a minimal amount of methanol, subsequently the amine (1.70 eq.) was added and the resulting mixture was stirred for 60 min over sodium sulfate for the imine for-mation (if both components were liquid, no methanol was needed unless the imine would precipitate). The perfluoro acid (1.00 eq.) was dissolved in a minimal amount of methanol and added to the imine at room temperature. The resulting mixture was stirred for 2 min. Subsequently, the isocyanide (1.70 eq.) was added to

the stirring mixture. The reaction was stirred for 1–3 d at room temperature. If a precipitate was formed during the reaction, additional methanol was added. If a long perfluorinated chain was present, tetrahydrofuran was utilized in order to homogenize the reaction mixture. The crude reaction mixture was dried under reduced pressure and purified via column chromatography (see General Purification Procedure: F-SPE). Typical yields: 20–85%.

*General procedure (GP2) for Ugi reactions: economic, small scale.* In a 10 mL round bottom flask, the aldehyde (1.30 eq., typically 500 μmol) was dissolved in a minimal amount of methanol. Subsequently, the amine (1.30 eq.) was added and the resulting mixture was stirred for 60 min over sodium sulfate. The perfluoro acid (1.00 eq.) was dissolved in a minimal amount of methanol and added to the imine at room temperature. The resulting mixture was stirred for 2 min. Subsequently, the isocyanide (1.20 eq.) was added to the stirring mixture. The reaction was stirred for 3 d at room temperature. The crude reaction mixture was dried under reduced pressure and purified via column chromatography (see General Purification Procedure: F-SPE section). Typical yields: 2%–35%. The lower yields compared to GP1 are most probably due a larger excess of components and higher concentrations favoring product formation for GP1.

### General purification procedures for F-SPE

For F-SPE separation, we suggest to utilize 1 g of perfluorinated silica gel per 50 mg of mixture subjected (5% by weight). However, in literature higher loadings of 5%–15% by weight are reported.

Step 1: Flush the perfluorinated silica gel with dimethylformamide (0.2 mL per g silica gel), gently applying positive pressure.

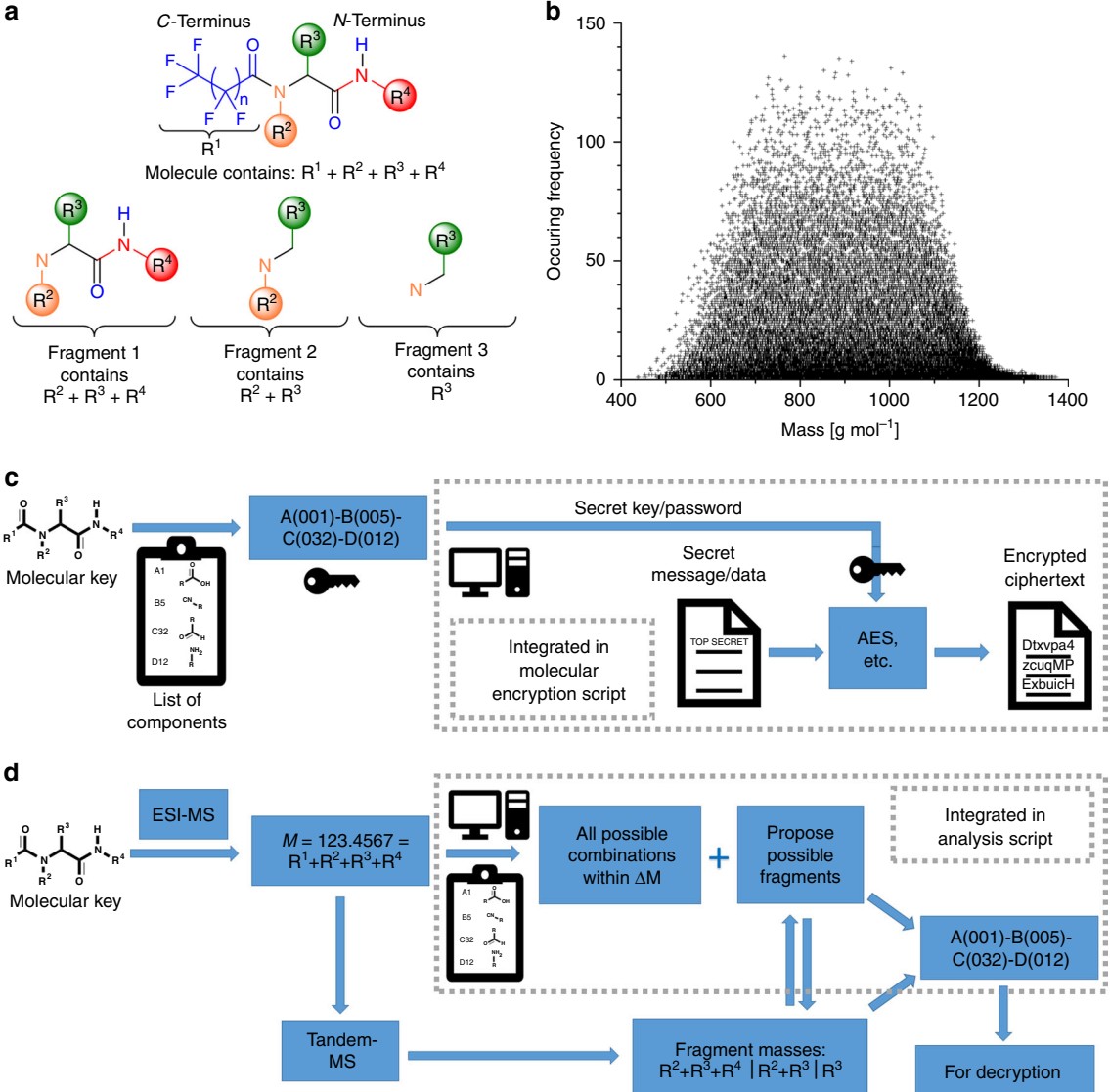

**Fig. 5** Cryptography integration of molecular keys. **a** Schematic representation of fragments, which can be used for the read out. Four masses are used for determining the sidechains $R^{1-4}$. **b** Database evaluation of the most probable masses occurring within a $\Delta M$ threshold of 0.01 Da. The possibilities are reduced from 500,000 to a maximum of 140 after entering the monoisotopic mass. **c** Schematic illustration of molecular encryption. Supplementary Software 2. **d** Schematic illustration of analysis script. Supplementary Software 1. A function explanation is included in the methods section.

Step 2: Precondition the column with methanol/water (8:2) (3 mL per g silica gel).

Step 3: Suspend the mixture to be purified in a minimal amount of dimethylformamide and load onto the preconditioned column.

Step 4: Elute the organic fraction with 2 mL methanol/water (80:20) per g silica gel. We suggest fractionating into tubes of appropriate size.

Step 5: Elute with 4 mL methanol per g silica gel to collect the fluorous fraction. Note: For long chain perfluorinated compounds, additionally elute with the same amount of tetrahydrofuran/acetone (1:1). Again, fractionating into tubes is desirable. TLC of the fractions indicates the complete elution of the desired compound.

Step 6: In order to recycle the F-SPE silica gel, flush with acetone and tetrahydrofuran and air dry. We do not suggest reusing the same silica gel more than three times.

After separation, the fractions were tested for purity via TLC, combined and concentrated under reduced pressure. If unreacted perfluoro acid or residual dimethylformamide were present in the fluorous fraction, a short silica gel filter column, eluting with *c*-hexane/ethyl acetate (3:1) was performed to yield the desired perfluorinated Ugi product.

Note: For purifying the molecular keys extracted from the various disguises 5 g of F-SPE silica gel were utilized.

**Hiding and isolating the molecular keys**. The molecular keys can be hidden in many creative ways. Since the molecular keys are flouro-tagged compounds, they can easily be isolated from various types of surrounding media. The main requirement to the surrounding environment or the matrix material is the absence of any perfluorinated alkyl chains. Furthermore, the extraction of the molecular key should be possible with reasonable effort.

The following extraction examples were performed with the molecular key A(005)-B(002)-C(004)-D(007).

Adsorbed onto paper: 4.0 mg of the molecular key were dissolved in 0.5 mL methanol and subsequently transferred dropwise onto the top right corner of an envelope. Intermediately, the drops were quickly evaporated by gently blow-drying with a heat gun. For further protection and disguising, the stamp was placed just above the covered area (see Supplementary Figure 3). For later extraction, the part of the paper doped with the molecular key (i.e., the area around the stamp) was cut into small pieces and extracted in an ultrasonic bath with 15 mL methanol for 15 min three times and subsequently three times with 10 mL dichloromethane. The extracts were combined, concentrated under reduced pressure, and purified via F-SPE. In this protocol 3.5 mg (90%) of the molecular key were recovered.

Dissolved in a perfume: 4.0 mg of the molecular key were dissolved in 0.5 mL ethanol and transferred into the same amount of a commercial perfume

(see Supplementary Figure 3). The resulting solution is more diluted than the initial perfume, but the dilution cannot be easily recognized by a change of appearance or smell. The molecular key proved too remain in solution for more than 14 days. It can be assumed, that the molecular key will not precipitate or phase separate (because a diluted system is used). For the extraction of the molecular key, the solution was evaporated under reduced pressure and subjected to F-SPE. In this protocol, 3.7 mg (93%) of the molecular key were recovered.

Adsorbed onto instant coffee powder, tea, or sugar: 4.0 mg of the molecular key were dissolved in 0.5 mL dichloromethane and added dropwise onto 1 g of the substrate powder, each drop was placed onto a different spot and allowed to evaporate separately. For the extraction, the powder was ground with a mortar and stirred three times in 5 mL dichloromethane for 3 min. The combined extracts were and filtered over celite® and washed with 10 mL water twice. The combined aqueous phases were reextracted with 5 mL dichloromethane twice. The combined organic phases were washed with 5 mL brine once, dried over sodium sulfate, evaporated under reduced pressure, and purified via F-SPE. Extraction yields: from instant coffee powder 3.4 mg (85%), from green tea 3.6 mg (90%), and from sugar 3.4 mg (87%), of the molecular key were recovered.

Note: For the extraction from sugar no further purification via F-SPE was necessary (fully water-soluble substrate).

From blood: 5.0 mg of the molecular key were dissolved in 0.5 mL ethanol and concentrated under reduced pressure until a transparent film of the molecular key with a minimized amount of ethanol was obtained. 5 mL of pig blood were added into the vial and gently stirred for 5 min. The blood should be stored and transported under cooling. For the extraction, the blood was diluted with 10 mL water and extracted with 20 mL dichloromethane three times (after the second extraction an emulsion was formed, hence the second and third organic extracts were combined and treated separately). The organic extracts were washed separately with 10 mL water twice. The organic phases were dried over sodium sulfate (emulsion broke upon drying), evaporated under reduced pressure, and purified via F-SPE. In this protocol, 4.5 mg (91%) of the molecular key were recovered.

Note: The extraction from blood was intended to demonstrate a challenging extraction example for the herein presented F-tagged molecular keys from complex biological media. Extraction from biological media would be far more challenging for cryptographic keys encoded in DNA or other biomolecules. The authors also want to clearly state that it is unethical to transport molecular keys in living organisms.

For MS analysis, even smaller amounts of substance are required so in principle a lower concentration of the molecular key could be employed. However, the herein obtained mass of several milligrams were sufficient for conduct TLC, GC-MS and NMR experiments for purity determination. GC-MS evidenced a purity of 99%, see Supplementary Figure 5, further indicating successful extraction and purification. After the molecular key has been isolated and purified, (tandem-) MS-spectra are obtained and hence the molecular key can be decrypted.

**General extraction procedure**. If the molecular key is adsorbed on a solid support, we propose to extract with dichloromethane several times (at least three) in order to fully recover the molecular key. Subsequently, the dichloromethane extract is washed with water and purified via F-SPE. If the molecular key is transported in solution, we propose concentration, and liquid/liquid extraction with dichloromethane and water. The combined organic layers are dried prior to F-SPE.

**Analyzing the molecular keys via tandem-MS**. First, the molecular key needs to be located, isolated, and purified. Subsequently, ESI-MS provides the monoisotopic mass (full scan mode 200–2000 $m/z$), which is used in the computer assisted readout (analysis script, Supplementary Software 1) to determine the most probable structures. The predominant signals arise from the intact molecule + sodium: $[M + Na]^+$ and two molecules + sodium: $[2 M + Na]^+$ (see Supplementary Figure 6). The sodium was introduced by utilizing a doped solvent mixture during the ESI sample preparation.

The next step towards the analysis of a molecular key is to isolate the $[M + Na]^+$ $m/z$ peak and further fragment the molecule. It proved to be most efficient to start with mild conditions (low fragmentation energy) and successively move on to higher fragmentation energy levels (see Supplementary Figure 7). For the first evaluation, a spectrum with many fragment peaks occurring in a wide range of molar masses (high information density) should be chosen and regarded in detail (i.e., 30 eV in Supplementary Figure 7 provides a high information density, since many fragments can be observed and in addition the unfragmented molecule at 595 $m/z$ is still present). However, the chosen energy should be as low as possible in order to avoid the further fragmentation of fragments. Furthermore, the observation of heavier/larger fragment species is more likely for lower fragmentation energies. In Supplementary Figure 7 the respective spectrum at 30 eV is displayed, the corresponding fragment evaluation can be found in the table below. As demonstrated in the evaluation, it is possible to observe numerous fragments and hence the information required for structure elucidation from one single spectrum.

However, in some cases, two tandem-MS spectra at different fragmentation energies are necessary to observe enough relevant fragments. In Supplementary Figure 8, two tandem-MS spectra of the same species (731 $m/z$) recorded at 35 eV

(a) and at 50 eV (b) collision energy are displayed. The heavier fragments indicated by the diamond cluster and asterisk can only be observed in the 35 eV energy spectrum (a, expansion). The lighter fragments indicated by the black circle and left-pointing triangle can solely be observed in the 50 eV spectrum (b). The corresponding fragment assignment is depicted in the corresponding table of Supplementary Figure 8.

In addition, tandem-MS fragment analysis can distinguish between isomers (molecules with same mass but different structures). In Supplementary Figure 9a, two isomeric molecular keys are displayed. These isomers cannot be simply distinguished by regarding the molar mass of the intact molecule (right-pointing triangle) nor the isomer fragments. In fact, it is necessary to evaluate the lighter fragments (black circle and black square), which are unambiguously different. The respective spectra are displayed in Supplementary Figure 9b (isomer 2) and c + d (isomer 1). The spectrum of the other isomer 2 presented in b shows a different and unique fragmentation pattern, and can therefore be distinguished from 1 (c + d). The fragment analysis of both spectra is presented in the table of Supplementary Figure 9. In conclusion, isomeric molecular keys can be distinguished in a straightforward fashion, if the list of components does not include isomers for the same component (as stated in the Criteria for the component database).

**Computer assisted readout for tandem-MS spectra**. The analysis of tandem-MS spectra at different energies turned out to be time consuming. At this point, a computer script (Supplementary Software 1) becomes a very useful tool for providing pre-calculated masses of the most probable fragments after the input of the monoisotopic mass of the respective molecular key. The mono isotopic mass can be obtained from ESI-MS spectra scanning in a range from 200 to 2000 $m/z$ (see Supplementary Figure 6). The script operates as follows: The database of components is the basis for the alphanumerical coding (e.g., aldehydes → letter A; benzaldehyde → A(003), etc.), the chemical formula, the corresponding exact masses and a SMILES code for visualizing the molecule, e.g., A(003) | benzaldehyde| $C_7H_6O$|106.04186|[[H]C(C1 = CC = CC = C1) = O.

The script then searches for all possible permutations of component combinations within the entered mass range (reduction of possibilities is illustrated in Supplementary Figure 4). The mono isotopic mass $[M + Na]^+$ is entered with four decimals, e.g., 567.5678 and an appropriate $\Delta M$ threshold (for the herein presented examples $\Delta M = 0.02$ Da is sufficient). The resulting possibilities are listed by the script and the masses of probable fragments are directly displayed. After entering additional fragments, the resulting possibilities are further refined until the structure is determined. The fragment masses are entered with two decimals (e.g., 123.45). If accidentally a wrong number is entered, the REDO command will return to the initial selection of possibilities after entering the $[M + Na]^+$ mass. After entering enough fragments, the selection of possibilities is reduced to one distinct molecule. With the alphanumerical information of the key [e.g., A(005)-B(002)-C(004)-D(007)], the hidden message can be decrypted. The script also offers the possibility to display more detailed information about the molecular key via the PRINT command. After selecting the print command a SMILES code for the target molecule and all precursor components (on basis of the list of components) are displayed. The SIMLES code can be copied, e.g., to ChemDraw® (insert with Alt + Crtl + P) for visualization of the molecular key structure (also refer to Supplementary Methods).

**Computer assisted encryption and decryption**. To supply a full toolchain, also a script is provided (Supplementary Software 2) that performs the actual encryption of messages (and the reverse process, the decryption of ciphertexts). The script operates either in encryption, or in decryption mode. In encryption mode, the script takes as input a plaintext (either as a file on disk, or entered on screen by the user), and a sequence of molecular keys. The molecular keys are then concatenated and used to derive an AES key for encryption, and the encrypted ciphertext is output either to a file, or onto the screen. In decryption mode, a ciphertext and molecular keys are entered, and the plaintext resulting from decryption is output.

Internally, the script uses the established OpenSSL tool to perform cryptographic operations (and in particular the key derivation and encryption/decryption operations). The main task of our script is hence to collect user input, and run the OpenSSL tool with the appropriate input and parameters. Specifically, the script chooses to encrypt/decrypt with the standard AES algorithm with 128-bit keys (that are derived using a standard cryptographic key derivation function from the molecular keys).

For decrypting the encrypted messages included as Supplementary Data 2, select decryption mode, drag, and drop the respective file into the window an enter the corresponding molecular key (obtained from the analysis script, Supplementary Software 1). Please export the messages in the format < filename > .txt (also refer to Supplementary Methods). For decrypting the encrypted file container please export in the format < filename > .zip. The solutions can be found in Supplementary Note 1.

**Purity determination**. The purity of the isolated molecular key [A(005)-B(002)-C(004)-D(007)] was further confirmed by GC-MS (Supplementary Figure 5). The intense signal at 11.5 min retention time corresponds to the displayed molecular key, the integral is 99%. The small impurity (1%) at 10.6 min retention time originates form a species with a shorter perfluorinated sidechain (5 $CF_2$ instead of 7 $CF_2$), which was incorporated because a shorter perfluorinated acid

component was present in the precursor material (ordered and used as received). The respective mass spectrum and the assignment of the fragments is presented in Supplementary Figure 5. However, since the amount of this impurity is very small we did not observe interference with other analytical methods or the tandem-MS readout.

**NMR characterization**. All substances synthesized in the context of this work were characterized via 1D and 2D NMR. Representative spectra for a molecular key [A(005)-B(002)-C(004)-D(007)] are displayed in Supplementary Figure 2. Most molecular keys showed a signal splitting due to restricted rotation (analogous to peptides), which was confirmed via NOESY spectroscopy (see Supplementary Figure 2c). In the $^1$H NMR spectrum (a) split signals can be observed for the protons on position CH$^1$ and NH$^6$. The signal between 5 and 6 ppm with a relative integral of two does not couple to other protons, as indicated in the COSY experiment (b). The HSQC experiment (e, displaying all carbon bond protons via $^1J_{(C-H)}$ correlations) indicates two cross signals and hence two carbon bound species (CH$^1$) in the respective area. The HMBC spectrum (f, displays correlations between carbons and protons that are separated by two, three, and sometimes even four bonds, i.e., in conjugated systems) was utilized to further confirm the molecular structure. The $^{19}$F stacked spectrum g of the molecular key (bottom of g) displays two new AB signals (originating from the CF$_2$$^{14}$ group next to the newly formed amide bond), if compared to the precursor.

**Data availability**. All relevant data is included as supplementary information and is also available from the corresponding author.

Availability of computer codes: All relevant scripts are included as supplementary information. The analysis script is provided in Supplementary Software 1, the encryption script is provided in Supplementary Software 2. Examples of encrypted messages and an encrypted file container are included in Supplementary Data 2.

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

## Acknowledgements

We thank PD Dr. Weiss, T. Neck, and Dr. N. Boukis for the discussions and comments on early versions of this manuscript. A.B. is grateful for the Chemie Fonds fellowship from the VCI. This work was financially supported in part by SFB 1176 (Projects A3 and Q5). We thank Prof. Podlech for sharing lab space with us. We acknowledge support by Deutsche Forschungsgemeinschaft and OpenAccess Publishing Fund of Karlsruhe Institute of Technology.

## Author contributions

A.B. and M.M. conceived and designed the project. A.B. designed the experiments with input from M.M. K.R. programed the analysis script. M.F. synthesized the molecular keys under the supervision of A.B. D.H. optimized the cryptography integration and programed the molecular encryption script. A.B. analyzed data, prepared the figures and wrote the paper, with feedback from all the authors.

## Additional information

**Competing interests:** The authors declare no competing interests.

