## [Peer Review File · Nature Communications]

Reviewers' comments:

Reviewer #1 (Remarks to the Author):

This is an interesting paper that focuses on a relatively underexplored research area: chemical-based cryptography (and steganography). The main novelty this paper using the Ugi reaction to generate a large number of molecular keys, as well the use of mass spectrometry (combined with AES) in data encryption decryption. The fairly simple separation of the fluorinated keys is another unique aspect of the semi-molecular encryption system. Indeed, these new features and ideas contribute to the emerging field of molecular cryptography. However, because this contribution is rather narrow and because this approach also suffers from notable limitations, I recommend publishing this work in a more specified journal after the following comments are addressed.

Comments

A key weakness of this paper is the form of presentation. Rather than focusing on the specific contribution to molecular cryptography, this paper unintentionally "oversells" the results. This, unfortunately, undermines the real achievements of this work. This reviewer believes that describing the achievements in the right context, and even mentioning limitations/future improvements, will actually strengthen the paper and highlight its originality.

The following comments are intended to improve the manuscript:

1. One claim mentions in this paper is that "a major weak point of cryptographic systems is the key itself, i.e. the (electronic) storage, or the transportation/distribution of the key".

Indeed, physical distribution of keys was a concern in earlier cryptographic system. Today, however, electronic communication is almost perfectly secured with combination of private and public key (e.g., RSA system). When sending a secured email, do we really need to coordinate the distribution of a new key to each receiver? In general, physical distribution of keys is not the preferred cryptographic tool. Modern cryptographic systems that use symmetric-key algorithms avoid the need for physical delivery of keys by using public-key algorithms (e.g., Diffie-Hellman). A valid argument is that a chemical approach offers a non-electronic communication channel, which could prevent spying on messages by hacking to electronic computers. This idea, however, namely, using a non-electronic chemical system that combines encryption with steganography, was recently discussed and demonstrated (reference 10). The possibility of hiding the keys on a paper and sending them by regular mail, as well as a way to avoid key distribution by using commercially available chemical keys, have also been demonstrated.

2. Another claim is that "Most of these systems employ complex molecules, synthesized in multistep procedures, for steganography. For a higher degree of security, steganography and cryptography can be combined." This argument is very vague. The systems discussed in the previous paragraph are different in their operation and purposes. Some are relatively simple to make. Others are very simple to operate. Applying the Ugi reaction for preparing libraries of keys is a very nice and interesting concept, which offers a new and a simple means to generate diverse keys. However, this method still requires compound purification, complex instrumentation, key distribution, and a non-trivial analysis by ES-MS. In addition, as noted by the authors, there are large libraries of chemical compounds that are commercially available. The Ugi reaction provides several specific advantages; but not an ultimate solution.

3. The following argument that "In contrast to the already described systems, we report a secret communication channel based on novel molecular keys, which can be easily hidden, transferred and isolated non-digitally, with the aim to significantly improve communication security." is not entirely accurate (e.g., reference 10).

4. "DNA, in comparison, has the drawbacks of replication/amplification protocols like the polymerase chain reaction, which can be undesirable for encryption keys." Without primers DNA cannot be amplified. Solid phase peptide synthesis (SPPS) and, in particular, 'split and pull/mix' approaches can also be used to obtain large libraries. The power of this work is in demonstrating a technique to generate large libraries of keys for cryptography, but not in the method itself.

5. The encryption protocol is not well described. Too many figures and examples (zip files, etc.) were moved to the SI section, which makes it difficult for the reader to follow. In addition, the authors combine chemical keys with AES, but do not elaborate on the AES method (except for

mentioning its applications). As a result, AES seems like a "black box" that does some magic. This reviewer is familiar with AES; however, the authors should explain to non-professional readers what the function of the "key" in AES is. The author should also explain "presedorandom" shown in the figure and make a clear distinction between cipher machines/encryption algorithm (AES) and molecular keys. Overall, the Ugi compounds do not encrypt the messages, but provide some random digits/bits.

6. In the main text, the author should show a real numeric example of the conversion of a particular compound (key) into a 'mass' and then, into a "numeric key", and then, into a very short cipher text. The "alphanumeric code" and other parameters discussed in the paper should be clearly defined and presented. I am not sure an "alphanumeric code" is the right definition in this work.

Reviewer #2 (Remarks to the Author):

The manuscript "Molecules for secret communication: Multicomponent reactions are the key" by Meier and co-workers describes the complementary use of cryptography (AES) and steganography using molecules as encryption keys. It uses multicomponent Ugi reaction concept to generate a virtual library of 500 000 molecules (some of them are also synthesized as examples) and then shows how these molecules can be transported (in solid or liquid media), extracted and analyzed by the intended recipient of the coded message. A software package that enables decoding and translation of the molecule (mass spectrum) to a key is also provided.

In general, I am very positive about this paper and recommend the publication after minor revisions. The Ugi reaction is well chosen and described, the use of perfluorinated chains is smart as it enables easy purification and the whole concept is really appealing. The presented data are convincing and fully support the hypothesis.

In my review, I focused on the chemistry/synthesis/analysis aspects due to my expertise. My comments are as follows:

1. The synthetic experimental procedures are written down with due diligence (my compliments!). However, original NMR spectra are missing for the synthesized compounds. Adding those to the SI is a commonly accepted practice in organic synthesis and would allow the present reviewer to better judge the quality and purity of the materials. Examples provided e.g. in SI figure 9 are too small for me to read them. Therefore I cannot well judge if the additional signals in the ¹H spectrum around the signal of proton #8 are impurities, residual solvents, or just my bad vision. I strongly encourage the Authors to add all the original ¹H and ¹³C spectra for the compounds described in SI.

2. My second criticism concerns the choice of compounds. My recommendation would be to choose those that, under given Ugi conditions, can form only one well defined product. The Authors take this into consideration (Supplementary Figure 1c) but seem to forget that there is a stereocenter formed in the Ugi product. Therefore, the use of a chiral substrate (like aldehydes 22 and 42; amines 6, 10, 12, 28, 29, 32, 41; isocyanide 6), would lead to emergence of diastereoisomeric mixtures which are difficult to separate and may give different MS fragmentations, complicating the picture. Could the Authors comment on that and also study this by including in the SI a synthesis starting from their chiral (I presume racemic) starting materials. My suggestion would be aldehyde 42, amine 12 and isocyanide 6; the choice of the acid I leave to the Authors).

3. In SI, please do not report two decimals for ¹³C spectra, one is enough.

4. Line 134 – why is this undesirable for encryption keys? A thermocycler is cheaper than tandem MS.

5. The use of blood as a carrier – could the authors explain the operational point of this, i.e. why anyone in real life would use blood (inherently difficult to transport, other than in the body of a human/animal) to transfer the decryption key? Any ideas this reviewer can come up with are scaring the living daylights out of him.

6. The paper misses a summary paragraph. In fact, the last paragraph is still describing methods and then shifts into some summary words, giving the paper an abrupt ending.

7. Minor: there is a typo in lines 66 (ore)

Reviewer #3 (Remarks to the Author):

The authors describe the use of MCR (Ugi reaction) to generate keys for the conversion of plaintext into ciphertext and vice versa. The enormous chemical space and diversity that can be created with MCR seems indeed interesting for the outlined purposes. Further, the use of steganography for concealing the key is discussed and demonstrated in some examples. The use of fluororous chemistry is well thought and assists to overcome possible complications in the separation of the key from other organic material. However, the basic idea of using chemistry to encrypt and conceal information is very close to an earlier publication by the Margulies group (ref. 10). In comparison the present work adds few new aspects, except from the different chemical approach. The analytical approach of identifying the key requires more sophisticated instrumentation as compared to Margulies' simpler fluorescence reading. In addition the multichromophoric approach by Margulies bears a non-linear response function, making it difficult to back engineer the key. In the present example back engineering would be possible by a combination of well established analytical methods (NMR, MS).

Altogether this is an interesting work, but too close to the previous publication by the Margulies group. Hence, I can not recommend the manuscript for publication in Nature Communications.

Reviewer #4 (Remarks to the Author):

Increasing the encoding capacity by employing a one-pot 4-component reaction is clever and original, especially because many examples of each component are available to mix and match. The use of several keys simultaneously but in different places further increases the encoding capacity.

The use of perfluoro groups in one component allows easy recovery of the molecular key from any non-fluorous matrix (by fluororous solid phase extraction) which represents most common media. The concealment aspect is therefore very effective. The tandem mass spectrometric readout is also effective because of its sensitivity.

There is one prior paper (Still WC, PNAS 1993,90, 10922) where fluorinated coding tags are analysed as barcodes via their gas chromatograms with fluorine-sensitive detectors. This paper deserves to be discussed and cited for another good use of fluorinated compounds in this general field.

Line 53; Steganography and cryptography have been combined previously in ref. 10. This should be cited here.

Line 55; the use of a nondigital step to increase security is discussed in ref 10. This should be cited here.

Line 66; not realized or even neglected

Line 176; the first molecular key

Smuggling a pun into the title is clever too.

This paper, following the minor revisions noted above, would be a nice addition to the growing field of molecular information processing by showing where molecular methods can solve problems which semiconductor and brute force methods cannot.

Answers to reviewer comments:

Use of colors:

black: original reviewer comments

green: response from the authors

highlighted in yellow: Quotations of changes in the manuscript

We would like to thank all reviewers for their time and constructive remarks, which certainly helped to improve the quality of this manuscript.

Reviewer #1 (Remarks to the Author):

This is an interesting paper that focuses on a relatively underexplored research area: chemical-based cryptography (and steganography). The main novelty this paper using the Ugi reaction to generate a large number of molecular keys, as well the use of mass spectrometry (combined with AES) in data encryption decryption. The fairly simple separation of the fluorinated keys is another unique aspect of the semi-molecular encryption system. Indeed, these new features and ideas contribute to the emerging field of molecular cryptography. However, because this contribution is rather narrow and because this approach also suffers from notable limitations, I recommend publishing this work in a more specified journal after the following comments are addressed.

Comments

A key weakness of this paper is the form of presentation. Rather than focusing on the specific contribution to molecular cryptography, this paper unintentionally “oversells” the results. This, unfortunately, undermines the real achievements of this work. This reviewer believes that describing the achievements in the right context, and even mentioning limitations/future improvements, will actually strengthen the paper and highlight its originality.

Answer: We have to apologize if we made this impression, but it was not our intention to “oversell”. We appreciate your criticism, but please also consider that it is very difficult to draw an objective

line in the use of language between “overselling” and highlighting. We carefully considered your comments and revised the manuscript throughout by paraphrasing and removing selected adjectives (e.g. very useful → useful, etc.) and believe this improves the manuscript as you suggested. Furthermore, we agree that a discussion of limitations is beneficial for the manuscript and thus added respective content in the introduction as well as in a short concluding paragraph.

The following comments are intended to improve the manuscript:

1. One claim mentions in this paper is that “a major weak point of cryptographic systems is the key itself, i.e. the (electronic) storage, or the transportation/distribution of the key”. Indeed, physical distribution of keys was a concern in earlier cryptographic system. Today, however, electronic communication is almost perfectly secured with combination of private and public key (e.g., RSA system).

Answer: We offer a qualitatively different method/channel to transport keys. A traditional key distribution mechanism is insecure against computationally unbounded adversaries, and in particular most used systems (e.g. with RSA or Diffie-Hellman used as key encapsulation/key agreement mechanisms) are not secure against (efficient) quantum adversaries. In contrast, our system offers a steganographic channel to transport keys (which, ideally, an adversary does not even notice). However, it is true that we do not communicate this clearly enough in the manuscript; thank you for pointing this out. We have changed the relevant portions of the introduction to (hopefully) make this clearer.

Furthermore, we changed the introduction according to your suggestions and included a discussion about RSA and Diffie-Hellman while paraphrasing the sentence: “a major weak point of cryptographic systems is the key itself, i.e. the (electronic) storage, or the transportation/distribution of the key”.

When sending a secured email, do we really need to coordinate the distribution of a new key to each receiver?

Answer: Our system is not intended as a substitute for email communication (or, say, securing internet connections). Indeed, the preparations necessary for securely transmitting a single key would seem to prohibit a large-scale use. Instead, we offer a highly robust physical steganographic

channel that is suitable for high-security applications. Again, we should have made this clearer (and we have changed the relevant parts of the revised manuscript accordingly).

In general, physical distribution of keys is not the preferred cryptographic tool.

Answer: For most use cases, we do agree (see also discussion above). In certain extreme cases (e.g. when using an asymmetric cryptography should be avoided due to its reliance on algebraic structures, while using symmetric encryption such as AES is still tolerable) however, a physical steganographic channel (as provided by our system) can provide a substitute for asymmetric cryptography. But we agree that in most use cases, using asymmetric cryptography, will be much more convenient.

Modern cryptographic systems that use symmetric-key algorithms avoid the need for physical delivery of keys by using public-key algorithms (e.g., Diffie–Hellman).

Answer: We are aware of RSA and Diffie-Hellman (and the concept of hybrid encryption). Please see above for a more detailed discussion. We have tried to clarify this in the paper.

A valid argument is that a chemical approach offers a non-electronic communication channel, which could prevent spying on messages by hacking to electronic computers. This idea, however, namely, using a non-electronic chemical system that combines encryption with steganography, was recently discussed and demonstrated (reference 10). The possibility of hiding the keys on a paper and sending them by regular mail, as well as a way to avoid key distribution by using commercially available chemical keys, have also been demonstrated.

Answer: Thank you for your comment. Indeed, also other reviewers mentioned that the differentiation from Margulies *et al.* (reference 10) should be made clearer. We highly appreciate the work from Margulies *et al.* and were inspired by the creative idea of hiding chemicals underneath the stamp of a letter for steganography. Firstly, we cited Margulies *et al.* more frequently throughout our manuscript at appropriate positions (as also suggested by other reviews). Moreover, in the course of this revision, we added a detailed description of differences to the concept of Margulies *et al.* in order to allow for a better differentiation.

The publication of Margulies *et al.* offers molecular messaging sensors designed to respond to various chemical inputs. The herein presented molecular keys encode the cryptographic key in their molecular structure and do not require additional chemical inputs. Please note that, if the molecular messaging sensors from Margulies *et al.* is contaminated with only one of the several (unintended) chemical inputs, the decoding strategy can be influenced drastically. Since the molecular sensor is very sensitive to various chemicals, the purity of the chemical input must be high or in the case of commercial products – fully reproducible. In this context, we want to point out that many commercial products are produced in batches and the exact composition can vary.

The herein presented molecular keys however are specifically designed for straightforward isolation and purification, thus overcoming the challenge associated with contaminations possibly interfering with the later readout. Additionally, the herein utilized molecular keys can be disguised in various environments and simply isolated even from complex media, enabling robust and effective steganography. The work of Margulies *et al.* employs a straightforward readout, but if compared to our approach, a significantly more demanding synthesis *via* a multi-step synthetic procedure is required to obtain the molecular sensors. In our system, we claim that each key is individual, and that the information stored within can be retrieved unambiguously. Margulies *et al.* present a linear discriminant analysis and claim 97% accuracy, but do not discuss the overall cryptographic capacity in terms of “bits”. Moreover, cross sensitivities can cause the same UV response under different conditions/different chemical input and hence influence the readout. On the contrary, our molecular keys in combination with the tandem-MS readout system allows an unambiguous identification of the key and hence a distinct assignment. Our manuscript clearly states the storage capacity of each molecular key and how the permutations are created. In addition, our molecular keys were fully integrated to AES encryption including a software package for readout and cryptography with specific examples given – in the context of molecular cryptography this is pioneering work – chemistry and cryptography have not been combined in such an extend previously.

In the manuscript, we changed the following passage accordingly:

“In contrast to already described systems (including the inspiring work of Margulies *et al.*¹⁰), we report a secret communication channel based on novel molecular keys, which can be easily hidden in various media, transferred nondigitally, isolated *via* F-tags and unambitiously read out *via* ESI-MS/MS (one of the most developed and sensitive analytical techniques).”

In summary, we strongly believe that the present work provides a valuable addition to the growing field of molecular cryptography and hope these answers also convince the reviewer and clarify the advantages/differences of the discussed systems. Ultimately, a potential user has to choose between a variety of available systems or even combine different methods.

2. Another claim is that “Most of these systems employ complex molecules, synthesized in multistep procedures, for steganography. For a higher degree of security, steganography and cryptography can be combined.” This argument is very vague. The systems discussed in the previous paragraph are different in their operation and purposes. Some are relatively simple to make. Others are very simple to operate. Applying the Ugi reaction for preparing libraries of keys is a very nice and interesting concept, which offers a new and a simple means to generate diverse keys. However, this method still requires compound purification, complex instrumentation, key distribution, and a non-trivial analysis by ES-MS.

Answer: We fully agree that the previously discussed approaches are unique and have their distinct benefits. The argument that “For a higher degree of security, steganography and cryptography can be combined” is in our opinion valid, but probably not accurate enough. We paraphrased this passage in our introduction accordingly.

“For a higher degree of security, decryption keys can be concealed by steganography *i.e.* via chemicals.”

To the best of our knowledge, up to date no contribution wherein molecular keys were specifically designed for isolation and MS-analysis are discussed. The statement that compound purification is required only applies if the purity of the obtained material is not sufficient to conduct further mass spectrometric analysis and therefore highly depends on the surrounding matrix wherein the molecular key was transported. For instance, if the molecular key was adsorbed on sugar no further purification *via* F-SPE was required (see methods section). However, the isolation protocol presented herein is fast, robust and efficient, allowing transportation of the molecular keys in various media.

Regarding your statement of “complex instrumentation”, we agree that high resolution mass spectrometers (further allowing collision experiments) are amongst the most developed analytical instruments. However, once the instrument is installed and operational, sample preparation and the subsequent measurements are straightforward and simple to operate for a trained person. In combination with the herein provided analysis script, the molecular keys can be analyzed with minimal effort. The whole readout procedure including sample preparation requires only a few

minutes for one molecular key. Please also consider that mass spectrometry is amongst the most sensitive analytical methods available, allowing trace amounts of molecular keys to be transported and analyzed. We added a note to the manuscript: ... “ESI-MS/MS (one of the most developed and sensitive analytical techniques)” ...

In addition, as noted by the authors, there are large libraries of chemical compounds that are commercially available. The Ugi reaction provides several specific advantages; but not an ultimate solution.

Answer: Although a different combinatorial approach could potentially be applied, the Ugi reaction was selected for the reasons provided in the manuscript (chapter: The molecular key: Design, synthesis and use), but we never claimed this to be an ultimate solution. After careful investigation and evaluation of different options in the beginning of this project, we could not determine a better or simpler solution than the Ugi reaction. Most importantly, a commercial library of millions of compounds would not allow a straightforward isolation (only very few compounds would be F-tagged, if at all) and the fragmentation pattern in MS/MS experiments would be different for each compound, thus preventing the use analysis scripts and making the key identification significantly more difficult, if not impossible. Our systematic approach allows to store a certain amount of information in each molecular key and can of course inspire scientists from different disciplines to invent other combinatorial approaches in the future. Please consider in this context that a systematic assignment of sidechains to alphanumeric codes was not discussed earlier in the concept of cryptography.

3. The following argument that “In contrast to the already described systems, we report a secret communication channel based on novel molecular keys, which can be easily hidden, transferred and isolated non-digitally, with the aim to significantly improve communication security.” is not entirely accurate (e.g., reference 10).

Answer: In our opinion, the claim that “we report a secret communication channel based on novel molecular keys” is valid. However, to clarify the difference to Ref. 10, we changed this sentence accordingly (please refer to the answers provided above).

4. “DNA, in comparison, has the drawbacks of replication/amplification protocols like the polymerase chain reaction, which can be undesirable for encryption keys.” Without primers DNA cannot be amplified. Solid phase peptide synthesis (SPPS) and, in particular, ‘split and pull/mix’ approaches can also be used to obtain large libraries.

Answer: Our intention was to state that DNA amplification is well established within forensic science and this technology could be employed to restore an allegedly destroyed cryptographic key. However, since the sentence is obviously leading to misunderstandings (also by other reviewers), we removed it from the manuscript.

The power of this work is in demonstrating a technique to generate large libraries of keys for cryptography, but not in the method itself.

Answer: The generation of a library of molecular keys is an important element of our system, but furthermore the system offers additional benefits: *i)* the list of components is flexible and adaptable; *ii)* the keys are easy to synthesize, chemically stable due to two amide bonds and designed for straight forward isolation utilizing F-tags; *iii)* the keys can be transported and separated effectively – even from complex biological media which might be challenging for other chemical steganography systems such as DNA and peptides (or the system reported in reference 10).

5. The encryption protocol is not well described. Too many figures and examples (zip files, etc.) were moved to the SI section, which makes it difficult for the reader to follow. In addition, the authors combine chemical keys with AES, but do not elaborate on the AES method (except for mentioning its applications). As a result, AES seems like a “black box” that does some magic. This reviewer is familiar with AES; however, the authors should explain to non-professional readers what the function of the “key” in AES is.

Answer: We included a comprehensive explanation (right after introducing AES), and highlight the function and critical role of the secret key in AES. Since the word count of *Nature Communications* manuscripts is strictly limited, we could unfortunately not move more examples to the main manuscript; however, the interested reader will also easily find the required information in the methods section and the SI.

The author should also explain “pseudorandom” shown in the figure and make a clear distinction between cipher machines/encryption algorithm (AES) and molecular keys. Overall, the Ugi compounds do not encrypt the messages, but provide some random digits/bits.

Answer: A bitstring is “pseudorandom” if and only if it is not efficiently distinguishable from a truly random bitstring. Usually, long pseudorandom bitstrings are generated deterministically from short, truly random bitstrings, using a pseudorandom generator. Pseudorandom generators can be built from a variety of assumed-to-be-hard computational problems, e.g., from any secure symmetric encryption scheme that has a short secret key. The expression pseudorandom did so far only appear in *Figure 5c*, in the revised version we removed the box “pseudorandom” from this figure accordingly.

6. In the main text, the author should show a real numeric example of the conversion of a particular compound (key) into a ‘mass’ and then, into a “numeric key”, and then, into a very short cipher text.

Answer: We appreciated your idea and changed the manuscript as following (keeping the restricted number of words in mind, thus unfortunately not being able to include an exemplary ciphertext to the manuscript):

First, we included an explanation in footnote 53:

“53. In an exemplary scenario, the sender would synthesize one or several molecular keys and determine the alphanumeric code associated with the chemical structures according to the list of components. In a particular case, the components perfluoropentanoic acid, 4-methoxyphenylisocyanide, pentylamine and benzaldehyde were chosen and converted into A(001)-B(012)-C(007)-D(007). This alphanumeric code is utilized to encrypt a message or data. The recipient isolates and analyzes a molecular key via ESI-MS and determines a m/z of 595.1615 Da for $[M+Na]^+$. After entering this mass into the analysis script (included in the SI), the most probable molecules and their respective fragments are listed. Fragmenting the $m/z = 595$ Da species *via* ESI-MS/MS generates characteristic fragments at $m/z = 176.14$ Da and 106.06 Da, which are entered into the script and are sufficient information to identify the molecular structure of the molecular key (*Figure 4*). Using the list of components, the corresponding alphanumeric code can now be generated and utilized to decrypt the message/data.”

Moreover, the caption of *Figure 2* was amended as follows:

Figure 2 | Steganographic key distribution: a, The sender and recipient discuss details about their secret communication and exchange the list of components (assigns chemical information to alphanumerical symbols). b, from left to right: **The sender synthesizes one or several molecular keys and determines the alphanumerical code – associated with the chemical structures according to the list of components. The “alphanumerical code – numeric key” e.g. A(001)-B(012)-C(007)-D(007) is serving as encryption key.** The molecular key is concealed and ...

We would highly appreciate if the editor would grant us the additional space required to include the above-mentioned explanations.

The “alphanumeric code” and other parameters discussed in the paper should be clearly defined and presented. I am not sure an “alphanumeric code” is the right definition in this work

Answer: The term alphanumeric code is the preferred alternative in this context. By definition (<http://www.dictionary.com/browse/alphanumeric>) “alphanumeric” includes: letters, numbers and special characters. The expression code can be defined as “a system used for brevity or secrecy of communication, in which arbitrarily chosen words, letters, or symbols are assigned definite meanings” (<http://www.dictionary.com/browse/code>). Both links were accessed on 5th Jan 2018. We added a short definition after the first occurrence of the term “alphanumeric code” in the manuscript.

However, because this contribution is rather narrow and because this approach also suffers from notable limitations, I recommend publishing this work in a more specified journal after the following comments are addressed.

Answer: We hope that we could overcome your concerns with our revisions. The present work is highly interdisciplinary and will provide valuable input to the scientific community. *Nature communications* remains the best choice for contributions of this type in our opinion.

Reviewer #2 (Remarks to the Author):

The manuscript “Molecules for secret communication: Multicomponent reactions are the key” by Meier and co-workers describes the complementary use of cryptography (AES) and steganography using molecules as encryption keys. It uses multicomponent Ugi reaction concept to generate a virtual library of 500 000 molecules (some of them are also synthesized as examples) and then

shows how these molecules can be transported (in solid or liquid media), extracted and analyzed by the intended recipient of the coded message. A software package that enables decoding and translation of the molecule (mass spectrum) to a key is also provided.

In general, I am very positive about this paper and recommend the publication after minor revisions. The Ugi reaction is well chosen and described, the use of perfluorinated chains is smart as it enables easy purification and the whole concept is really appealing. The presented data are convincing and fully support the hypothesis.

In my review, I focused on the chemistry/synthesis/analysis aspects due to my expertise. My comments are as follows:

1. The synthetic experimental procedures are written down with due diligence (my compliments!).

However, original NMR spectra are missing for the synthesized compounds. Adding those to the SI is a commonly accepted practice in organic synthesis and would allow the present reviewer to better judge the quality and purity of the materials.

Examples provided e.g. in SI figure 9 are too small for me to read them. Therefore, I cannot well judge if the additional signals in the ^1H spectrum around the signal of proton #8 are impurities, residual solvents, or just my bad vision.

I strongly encourage the Authors to add all the original ^1H and ^{13}C spectra for the compounds described in SI.

Answer: Thank you for your recommendation. We included full spectroscopic proof (including ^1H , ^{13}C , ^{19}F and 2D experiments) for the identification of the herein synthesized substances in the SI.

Indeed, the size of the illustrations in *SI Figure 9* are small, but of high resolution thus enabling a zoom in within the electronic version. We therefore did not change *Figure 9* (and other figures) as the SI is already long.

Concerning your question about proton #8, this is quite interesting: The additional signals around the centroid peak are not caused by impurities but belong to the same molecule (evidenced by the fact that the proton signal on position #8 is bound to the same carbon atom – see Supplementary *Figure 9 e*: HSQC - blue cross signal for position #8 – or the additionally added spectra in the SI). This peak splitting (causing signals of higher order) is most probably the consequence of a combination of several effects:

i) restricted rotation (discussed in the text of *SI Figure 9*), which was confirmed *via* NOESY experiments, *SI Figure 9 c*).

ii) the enantiotopic environment of the protons on position #8 causing AB-signal splitting.

However, these split signals complicated the NMR spectra (also affecting the ^{13}C and ^{19}F spectra) and therefore we also included COSY, HSQC and HMBC spectra to enable the reader to follow our structural assignments. The HMBC spectra are further useful to clearly identify the quaternary carbon atoms in cases where the signal to noise of the respective ^{13}C spectra is weak in the quaternary region (whenever this was the case we added a small note in the respective figure caption).

The purity of the material presented in *SI Figure 9* is further illustrated and discussed in *SI Figure 8* (*via* gas chromatography).

2. My second criticism concerns the choice of compounds. My recommendation would be to choose those that, under given Ugi conditions, can form only one well defined product. The Authors take this into consideration (Supplementary Figure 1c) but seem to forget that there is a stereocenter formed in the Ugi product. Therefore, the use of a chiral substrate (like aldehydes 22 and 42; amines 6, 10, 12, 28, 29, 32, 41; isocyanide 6), would lead to emergence of diastereoisomeric mixtures which are difficult to separate and may give different MS fragmentations, complicating the picture.

Could the Authors comment on that and also study this by including in the SI a synthesis starting from their chiral (I presume racemic) starting materials. My suggestion would be aldehyde 42, amine 12 and isocyanide 6; the choice of the acid I leave to the Authors).

Answer: Thank you for this interesting remark. The F-SPE purification protocol retains perfluorinated compounds selectively and is thus unaffected by diastereoisomers. This is a great advantage of F-SPE and was utilized numerous times in so called fluororous mixture synthesis (FMS).¹ This FMS separation was also applied successfully for the synthesis of diastereomer mixtures, wherein F-tagged diastereomers were synthesized stepwise and separated from non-fluorinated contaminations *via* F-SPE.²

We did not forget about chirality in our considerations, but we were not concerned about diastereoisomer fragmentation because of our experience with MS analysis and previous literature on this topic. All diastereoisomers have the same molar mass and hence the first information required for the readout (*i.e.* $[\text{M}+\text{Na}]^+$) is the same for all diastereoisomers. If a diastereomeric

mixture is subjected to a tandem-MS experiment, it can be assumed that the required fragments for the readout will still be observed, since, even if theoretically different fragmentation pathways may occur, the favored fragments presented in the manuscript will still be observed, at least for some of the fragmentation pathways (maybe with a somewhat lower probability, but the required fragments should still be present and detectable). The observed mass of the fragments is independent from the stereochemical information. The fragmentation behavior of diastereoisomers is well known in MS/MS methods and was previously studied, confirming that the fragmentation leads to a different intensity distribution pattern but however similar fragments were observed.^{3,4}

In order to fully overcome your concerns, we performed a synthesis according to your suggestion (the starting materials you mentioned could unfortunately not be delivered in time and we thus chose available ones that still proof the concept), starting from, as you presumed, racemic components and resulting in a mixture of diastereoisomers:

The respective procedure along with analysis and displayed NMR data can be found in the SI (last entry). We also added a new paragraph to the SI named “**influence of stereochemistry**”, wherein the above-mentioned considerations are explained to the reader and the ESI-MS/MS spectra of the new compound are included. In the methods section, the following comment was added:

Diastereomeric mixtures of molecular keys synthesized from racemic starting components can be treated and utilized similar to stereomerically pure molecular keys (a study commenting on the influence of stereochemistry is included in the SI).

1. Luo, Z. ., Zhang, Q., Oderaotshi, Y. & Curran, D. P. Fluorous Mixture Synthesis: A Fluorous-Tagging Strategy for the Synthesis and Separation of Mixtures of Organic Compounds. *Science* **291**, 1766–1769 (2001).
2. Lu, Y. *et al.* Fluorous diastereomeric mixture synthesis (FDMS) of hydantoin-fused hexahydrochromeno[4,3-b]pyrroles. *Chem. Commun.* **46**, 7578 (2010).

3. Madhusudanan, K. P. Tandem mass spectra of ammonium adducts of monosaccharides: differentiation of diastereomers. *J. Mass Spectrom.* **41**, 1096–1104 (2006).
4. Drabik, E. *et al.* Differentiation of Diastereoisomers of Protected 1,2-Diaminoalkylphosphonic Acids by EI Mass Spectrometry and Density Functional Theory. *J. Am. Soc. Mass Spectrom.* **24**, 388–398 (2013).

3. In SI, please do not report two decimals for ^{13}C spectra, one is enough.

Answer: In fact, throughout literature, ^{13}C chemical shifts are mostly reported with one decimal. However, in our opinion, the accuracy of modern spectrometers allows signal differentiation in the range of 0.05 – 0.005 ppm for high resolution ^{13}C -NMR spectra. Clearly, the second decimal will be somewhat error-prone and should only be used for a relative distinction (i.e. if a peak is shifted more to high or low field). Nevertheless, we changed the ^{13}C chemical shifts to one decimal as suggested.

4. Line 134 – why is this undesirable for encryption keys? A thermocycler is cheaper than tandem MS.

LINE 139: DNA, in comparison, has the drawbacks of replication/amplification protocols like the polymerase chain reaction, which can be undesirable for encryption keys.

Answer: Our intention was to state that DNA amplification is well established within forensic science and this technology could be employed to restore an allegedly destroyed cryptographic key. However, since the sentence is obviously leading to misunderstandings (also by other reviewers), we removed it from the manuscript.

5. The use of blood as a carrier – could the authors explain the operational point of this, i.e. why anyone in real life would use blood (inherently difficult to transport, other than in the body of a human/animal) to transfer the decryption key? Any ideas this reviewer can come up with are scaring the living daylights out of him.

Answer: By hiding the molecular key in a blood sample, we intended to demonstrate that perfluorinated Ugi compounds can be extracted from complex biological media. Extraction from biological media would be far more challenging for other cryptographic keys encoded in DNA or other biomolecules. Of course, we did not want to make the impression that our keys can be used in living animals/humans. We added a note to clarify this:

“Note: The extraction from blood was intended to demonstrate a challenging extraction example for the herein presented F-tagged molecular keys from complex biological media. Extraction from biological media would be far more challenging for cryptographic keys encoded in DNA or other biomolecules. The authors also want to clearly state that it is unethical to transport molecular keys in living organisms.”

6. The paper misses a summary paragraph. In fact, the last paragraph is still describing methods and then shifts into some summary words, giving the paper an abrupt ending.

Answer: Thank you for this remark; since our word count is strictly limited we did not include a separate summary in the initially submitted version of the manuscript. However, we prepared a sample text that we would very much like to include at the end of our manuscript. We would highly appreciate if the editors would accept these concluding words and the corresponding marginally increased word count for our publication and follow this reviewer suggestion.

“Molecular keys with a data storage capacity of approximately 18 bit were synthesized in a convergent one-pot reaction approach. The molecular structures were analyzed by a combination of tandem-MS fragmentation and computer assisted readout (analysis script included in the SI). The respective structures served as decryption keys for AES encrypted messages or data files (cryptography scrip is included in the SI). In principle, the herein presented keys could also be used for other applications, e.g. as identification or anti counterfeit tags.”

7. Minor: there is a typo in lines 66 (ore)

Answer: Corrected.

Reviewer #3 (Remarks to the Author):

The authors describe the use of MCR (Ugi reaction) to generate keys for the conversion of plaintext into ciphertext and vice versa. The enormous chemical space and diversity that can be created with MCR seems indeed interesting for the outlined purposes. Further, the use of steganography for concealing the key is discussed and demonstrated in some examples. The use of fluororous chemistry

is well thought and assists to overcome possible complications in the separation of the key from other organic material.

However, the basic idea of using chemistry to encrypt and conceal information is very close to an earlier publication by the Margulies group (ref. 10). In comparison, the present work adds few new aspects, except from the different chemical approach.

Answer: We highly appreciate the work from Margulies *et al.* and were inspired by the creative idea of hiding chemicals underneath the stamp of a letter for steganography. In the course of this revision, we added a detailed description of differences to the publication of Margulies *et al.* in order to allow for a better differentiation and point out advantages/disadvantages.

Please especially see the detailed answers to comments from reviewer #1 (we thought it would be unpractical/not easy to read to copy-paste the complete answer also at this position).

In summary, we strongly believe that the present work provides a valuable addition to the growing field of molecular cryptography and hope these answers also convince the reviewer and clarify the advantages/differences of the discussed systems.

The analytical approach of identifying the key requires more sophisticated instrumentation as compared to Margulies' simpler fluorescence reading.

Answer: We agree that high resolution mass spectrometers (further allowing collision experiments) are amongst the most developed analytical instruments. However, once the instrument is installed and operational, sample preparation and the subsequent measurements are straightforward and simple to operate for a trained person. In combination with the herein provided analysis script the molecular keys can be analyzed with minimal effort. The whole readout procedure including sample preparation requires only a few minutes for one molecular key. Please also consider that mass spectrometry is amongst the most sensitive analytical methods available, allowing trace amounts of molecular keys to be transported and analyzed.

In addition the multichromophoric approach by Margulies bears a non-linear response function, making it difficult to back engineer the key. In the present example back engineering would be possible by a combination of well established analytical methods (NMR, MS).

Answer: In order to avoid back engineering of the herein established strategy, several hierarchies of security are included (see Figure 2c). Full structure elucidation without assumptions about the molecular structure is not feasible. Hence, an adversary needs a substantial amount of information in order to determine the molecular structure of a key. Only trace amounts of molecular key must be transported, thus the adversary has only a very limited number of attempts to isolate the key and determine the structure.

Altogether this is an interesting work, but too close to the previous publication by the Margulies group. Hence, I can not recommend the manuscript for publication in Nature Communications.

Answer: We hope that the details now provided here and especially in the detailed answer to reviewer #1 also can convince this reviewer that we have made very significant progress addressing critical issues of the mentioned previous publication.

Reviewer #4 (Remarks to the Author):

Increasing the encoding capacity by employing a one-pot 4-component reaction is clever and original, especially because many examples of each component are available to mix and match. The use of several keys simultaneously but in different places further increases the encoding capacity. The use of perfluoro groups in one component allows easy recovery of the molecular key from any non-fluorous matrix (by fluorous solid phase extraction) which represents most common media. The concealment aspect is therefore very effective. The tandem mass spectrometric readout is also effective because of its sensitivity.

There is one prior paper (Still WC, PNAS 1993,90, 10922) where fluorinated coding tags are analysed as barcodes via their gas chromatograms with fluorine-sensitive detectors. This paper deserves to be discussed and cited for another good use of fluorinated compounds in this general field.

Answer: Thank you very much for making us aware of this (only somewhat related) manuscript; we included the reference accordingly in the introduction:

... “molecular tags (equipped with halogen substituted aromatic sidechains) serving as barcodes for the identification of chemical libraries *via* gas chromatography.^[41] “

Line 53; Steganography and cryptography have been combined previously in ref. 10. This should be cited here.

Answer: Corrected, we cited accordingly.

Line 55; the use of a nondigital step to increase security is discussed in ref 10. This should be cited here.

Answer: Corrected, we cited accordingly.

Line 66; not realized or even neglected

Answer: Corrected.

Line 176; the first molecular key

Answer: Corrected.

Smuggling a pun into the title is clever too. This paper, following the minor revisions noted above, would be a nice addition to the growing field of molecular information processing by showing where molecular methods can solve problems which semiconductor and brute force methods cannot.

Reviewers' Comments:

Reviewer #1 (Remarks to the Author):

I will first begin with a positive note that this is a nice paper that should be published. In addition, I do appreciate the hard work that was required to develop the various experiments, and the time the authors spent on correcting the paper and writing the response letter. The paper has improved to some extent (see comments).

On the other hand, unfortunately, I cannot support publishing this work in Nature Communications. The reasons for the original rejection were not based my misunderstanding of the new aspects involving this technology, but rather on the fact that these points are not sufficiently novel for publication in this journal.

I will try to do a better job in explaining this issue in the following comments.

Comment 1:

Referee 3 indicated that the system is "very close to an earlier publication by the Margulies group (ref. 10)". Although in their response letter the authors provide a detailed explanation of the differences between the two systems (their Ugi reaction-MS system vs. the system presented in reference 10), I fully agree with referee 3 that these systems are just too close.

I hope the following points would better clarify why this paper should be published in another journal:

(1) In reference 10, small molecules that can be used to encrypt messages were also hidden on a paper (steganography), sent by regular mail, and were used to encrypt messages (cryptography) using fluorescence microscopy. Yes, there might be a few advantages in using MS / Ugi reaction/ and f- tags. These advancement, however, are not sufficiently novel for publication in this journal. There are also various disadvantages of the MS-Ugi method (which I am not going to discuss), but this is not the reason for rejection.

(2) Perhaps the authors have missed the point that in reference 10, not only a cipher molecular machine was hidden in a paper and sent by regular mail, but also the unique molecular inputs. Considering that in both manuscripts chemical inputs are hidden on a paper, the experiment shown in Figure 3 (the figure number in the caption should be corrected) becomes very similar to figure 5 in reference 10.

(3) The authors should be aware that in ref 10, not only chemical inputs were developed (in fact, this is a minor aspect of the system), but also a molecular cipher machine. Namely, a molecular machine was used to encrypt and decrypt the data. Here, the authors use molecular inputs but rely on the AES encryption, which requires the use of electronic computers.

(4) The idea of combining cryptography with steganography was clearly discussed in reference 10. This point should be specifically mentioned in the revised version. See comment 4.

Comment 2:

Following the comments by three referees (regarding the similarity between the systems), the revised version of the manuscript better refers the earlier work in reference 10.

However, the author must also revise the following sections:

(1) Line 61: "In contrast to the already described systems (including the inspiring work of Margulies et al. 10), we report a secret communication channel based on novel molecular keys, which can be easily hidden in various media, transferred nondigitally, isolated via F-tags and unambitiously read out via ESI-MS/MS (one of the most developed and sensitive analytical techniques)."

The term "in contrast" is misleading and should be replaced by other sentence. For example, by: "Herein we report a means to advance chemical communication systems (including the inspiring systems developed by of Margulies et al. 10)" (or an equivalent sentence).

The reason for removing the "in contrast" term is that: (i) "A secret communication channel based

on novel molecular keys" was demonstrated in ref 10, (ii) "which can be easily hidden" was demonstrated in ref 10, and (iii) "transferred nondigitally" was demonstrated and discussed in ref 10.

This paper shows some advances, but these advances are certainly is not "in contrast" to what has been achieved.

Comment 3:

I suggest avoiding the comparison to current electronic encryption techniques. I mentioned it in my previous response to highlight the fact that there are also various limitations in the UGI-MS-AES cryptography system. My intention was that the authors would present their results in the context of chemical cryptography-steganography (e.g., ref 10), but not that they would undermine the electronic approach. I take the blame for not explaining correctly, but strongly believe that this paragraph should be removed:

"...For instance, most of the asymmetric cryptographic schemes in use today (including the Diffie-Hellman and RSA schemes) can be broken with a (currently hypothetical) quantum computer.

This paragraph implies that the Ugi-MS system is better than the current computer encryption methods, which can only be cracked by futuristic quantum computers....

Comment 4

In the response letter that there are several points in which the author claim to address the referees' comment, but I couldn't find the relevant change in the text:

(1) Line 48: "For a higher degree of security, decryption keys can be concealed by steganography i.e. via chemicals."

Referee 4 requested that ref 10 should be cited here.

Although he author claim that "Corrected, we cited accordingly", this request was not followed.

(2) Line 63: "Easily hidden in various media, transferred nondigitally"

Referee 4 requested that ref 10 should be cited here.

Although he author claim that "Corrected, we cited accordingly", this request was not followed.

(3) In response to referee 1, the authors claim that "Furthermore, we agree that a discussion of limitations is beneficial for the manuscript and thus added respective content in the introduction as well as in a short concluding paragraph." No such discussion appears in the conclusion.

Reviewer #2:

Remarks to the Author:

I would like to thank the authors for addressing the points I raised in a very serious and dedicated way. I would also encourage teh Editor to allow the few extra lines of text for the conclusion (see reviewer 2, question 6).

Reviewer #3 (Remarks to the Author):

The revised version of the manuscript and more significantly the detailed authors' response to the reviewers' criticism have changed my initial view on the manuscript. I think the few but very precise re-phrasing/additions that the authors have included in the manuscript contrast their work against Margulies' work sufficiently. They are correct in saying that their key is buried in the chemical structure of straightforward generated compounds, instead of using supramolecular processes and/or the modulation of photophysical properties of fluorophores. This may indeed be an advantage in terms of a robust performance of the decryption procedure. Still it is instrumentally demanding and less flexible in terms of mobility. I think that the undertaken

revision and argumentation deserve a positive assessment and justify the acceptance of the manuscript.

Answers to reviewer comments:

Use of colors:

black: original reviewer comments

green: response from the authors

highlighted in yellow: Quotations of changes in the manuscript from revision 1

highlighted in red: Quotations of changes in the manuscript from revision 2

Revision 2:

We would like to thank all reviewers for their time and constructive remarks for the second review process.

Reviewer #1 (Remarks to the Author):

I will first begin with a positive note that this is a nice paper that should be published. In addition, I do appreciate the hard work that was required to develop the various experiments, and the time the authors spent on correcting the paper and writing the response letter. The paper has improved to some extent (see comments).

On the other hand, unfortunately, I cannot support publishing this work in Nature Communications. The reasons for the original rejection were not based my misunderstanding of the new aspects involving this technology, but rather on the fact that these points are not sufficiently novel for publication in this journal.

I will try to do a better job in explaining this issue in the following comments.

Comment 1:

Referee 3 indicated that the system is “very close to an earlier publication by the Margulies group (ref. 10)”. Although in their response letter the authors provide a detailed explanation of the differences between the two systems (their Ugi reaction-MS system vs. the system presented in reference 10), I fully agree with referee 3 that these systems are just too close. I hope the following points would better clarify why this paper should be published in another journal:

Answer: Dear Reviewer #1, please note that Reviewer #3 has changed his initial view on the manuscript is now of the opinion that the authors of the current manuscript “contrast their work against Margulies' work sufficiently.”

(1) In reference 10, small molecules that can be used to encrypt messages were also hidden on a paper (steganography), sent by regular mail, and were used to encrypt messages (cryptography) using fluorescence microscopy. Yes, there might be a few advantages in using MS / Ugi reaction/ and f- tags. These advancement, however, are not sufficiently novel for publication in this journal. There are also various disadvantages of the MS-Ugi method (which I am not going to discuss), but this is not the reason for rejection.

Answer: Indeed, we see some parallels to reference 10, but instead of focusing on what was reported previously in reference 10, we want to emphasize the differences and unique advantages of both contributions. In fact, reference 10 also used small molecules hidden on paper. This example was very well chosen because of the didactic value for the reader. However, this concept is not new to reference 10, because ever since the invention of secret ink, chemicals were hidden on paper, sent by mail and used to encrypt messages. However, as also explained in our previous reply letter, we were inspired by reference 10 (*i.e.* utilizing the concept of hiding chemicals on paper and sending by mail), because this concept can easily be understood and is therefore very well suited for explaining the steganography concept to the reader. Please also consider that our work utilized different examples such as hiding on coffee, tea, blood etc.

(2) Perhaps the authors have missed the point that in reference 10, not only a cipher molecular machine was hidden in a paper and sent by regular mail, but also the unique molecular inputs. Considering that in both manuscripts chemical inputs are hidden on a paper, the experiment shown in Figure 3 (the figure number in the caption should be corrected) becomes very similar to figure 5 in reference 10

Answer: We agree that both figures are somewhat similar, but please take into account that these figures are highly simplified schematic representations. Furthermore, please refer to our answer provided for (1) and the answers provides in revision 1 of this manuscript.

(3) The authors should be aware that in ref 10, not only chemical inputs were developed (in fact, this is a minor aspect of the system), but also a molecular cipher machine. Namely, a molecular machine was used to encrypt and decrypt the data. Here, the authors use molecular inputs but rely on the AES encryption, which requires the use of electronic computers.

Answer: Yes, we are aware of this and highly appreciate the concepts of reference 10. Our herein presented strategy intentionally includes external encryption schemes (such as e.g. AES) for greater flexibility, but of course at the cost of an additional external processing step. . In particular, please note that AES is only one example and that the molecular key concept can be used for every symmetric encryption scheme. As mentioned in the manuscript, a potential user has to evaluate pros and cons of a certain system and find the best suitable method for the anticipated application – and both manuscripts in question offer such pros and cons depending on the intended use.

(4) The idea of combining cryptography with steganography was clearly discussed in reference 10. This point should be specifically mentioned in the revised version. See comment 4.

Answer: Again, the contributions of reference 10 are very valuable for the scientific community and were inspiring for our own manuscript. However, combinations of steganography and cryptography are discussed in literature for many years and are therefore not unique to reference 10. Nevertheless, we again cited reference 10 to overcome all concerns (see comment 4) and also added further reference to earlier work.

1. Almuhammadi, S. & Al-Shaaby, A. A Survey on Recent Approaches Combining Cryptography and Steganography. in *Computer Science & Information Technology (CS & IT) 63–74* (Academy & Industry Research Collaboration Center (AIRCC), 2017). doi:10.5121/csit.2017.70306
2. Challita, K. & Farhat, H. Combining Steganography and Cryptography: New Directions. *Int. J. New Comput. Archit. Their Appl.* 1, 199–208 (2011).
3. Raphael, A. J. & Sundaram, V. Cryptography and steganography – A survey. *Int. J. Comput. Technol. Appl.* 2, 626–630 (2011).

Comment 2:

Following the comments by three referees (regarding the similarity between the systems), the revised version of the manuscript better refers the earlier work in reference 10.

However, the author must also revise the following sections:

(1) Line 61: “In contrast to the already described systems (including the inspiring work of Margulies et al. 10), we report a secret communication channel based on novel molecular keys, which can be easily hidden in various media, transferred nondigitally, isolated via F-tags and unambitiously read out via ESI-MS/MS (one of the most developed and sensitive analytical techniques).”

The term “in contrast” is misleading and should be replaced by other sentence. For example, by: “Herein we report a means to advance chemical communication systems (including the inspiring systems developed by of Margulies et al. 10)” (or an equivalent sentence).

The reason for removing the “in contrast” term is that: (i) “A secret communication channel based on novel molecular keys” was demonstrated in ref 10, (ii) “which can be easily hidden “was demonstrated in ref 10, and (iii) “transferred nondigitally” was demonstrated and discussed in ref 10.

This paper shows some advances, but these advances are certainly is not “in contrast” to what has been achieved.

Answer: We revised “in contrast to” according to your suggestions.

“Herein we report a means to advance chemical communication systems (including the inspiring work of Margulies *et al.*), *via* a secret communication channel based on novel molecular keys, which can be easily hidden in various media, transferred nondigitally,¹ isolated *via* F-tags and unambitiously read out *via* ESI-MS/MS (one of the most developed and sensitive analytical techniques). ~~nondigitally, with the aim to significantly improve communication security....”~~

Comment 3:

I suggest avoiding the comparison to current electronic encryption techniques. I mentioned it in my previous response to highlight the fact that there are also various limitations in the UGI-MS-AES cryptography system. My intention was that the authors would present their results in the context of chemical cryptography-steganography (e.g., ref 10), but not that they would undermine the electronic approach. I take the blame for not explaining correctly, but strongly believe that this paragraph should be removed:

“...For instance, most of the asymmetric cryptographic schemes in use today (including the Diffie-Hellman and RSA schemes) can be broken with a (currently hypothetical) quantum computer.

This paragraph implies that the Ugi-MS system is better than the current computer encryption methods, which can only be cracked by futuristic quantum computers....

Answer: Thank you for explaining your previous response in a clearer way. Even though our above-mentioned sentence is valid, we removed this sentence according to your suggestion in order to avoid misunderstandings.

Comment 4

In the response letter that there are several points in which the author claim to address the referees' comment, but I couldn't find the relevant change in the text:

(1) Line 48: "For a higher degree of security, decryption keys can be concealed by steganography i.e. via chemicals."

Referee 4 requested that ref 10 should be cited here.

Answer: We want to apologize for missing this point in the first revision and corrected accordingly. The initial reference 10 is now reference 1 in the manuscript, but was continued to be referred to as "reference 10" throughout our answers herein.

Although he author claim that "Corrected, we cited accordingly", this request was not followed.

(2) Line 63: "Easily hidden in various media, transferred nondigitally"

Referee 4 requested that ref 10 should be cited here.

Answer: Again, we want to apologize and corrected accordingly at the end of the paragraph (also see comment 2 (1)). Here, we want to point out that reference 10 is now cited three times throughout the manuscript, listed as reference 1 in our bibliography and mentioned as "inspiring work" for our study. We hope this will fully convince you, that the visibility of reference 10 is ensured in our manuscript and we really appreciate this work.

Although he author claim that "Corrected, we cited accordingly", this request was not followed.

(3) In response to referee 1, the authors claim that "Furthermore, we agree that a discussion of limitations is beneficial for the manuscript and thus added respective content in the introduction as well as in a short concluding paragraph." No such discussion appears in the conclusion.

Answer: The limitations of our work are included in several passages within the manuscript. The following passages were selected in order to justify our statement:

- (1) "Typically, secret keys are short (e.g. 128 bits) ... "
- (2) "Molecular keys with a data storage capacity of approximately 18 bit were synthesized in a one-pot reaction approach."

- (3) Data storage capacity is admittedly much higher for DNA than for synthetic molecules, which might change in the near future with the development of sequence defined macromolecules

The combination of arguments (1) – (3) leads to the conclusion that the storage capacity of one single key can be regarded as a limiting factor. However, this challenge was overcome in our work by utilizing several keys at once. Argument (3) further describes potential developments for future research.

To make this point even clearer in the manuscript, we added a sentence at the end of the conclusion.

"In future research, the investigation of molecular keys with even higher data storage capacities will be of great interest in order to overcome current limitations of the system."

Reviewer #2 (Remarks to the Author):

I would like to thank the authors for addressing the points I raised in a very serious and dedicated way. I would also encourage the Editor to allow the few extra lines of text for the conclusion (see reviewer 2, question 6).

Answer: Dear Reviewer #2, thank you very much for your positive feedback.

Reviewer #3 (Remarks to the Author):

The revised version of the manuscript and more significantly the detailed authors' response to the reviewers' criticism have changed my initial view on the manuscript. I think the few but very precise re-phrasing/additions that the authors have included in the manuscript contrast their work against Margulies' work sufficiently. They are correct in saying that their key is buried in the chemical structure of straightforward generated compounds, instead of using supramolecular processes and/or the modulation of photophysical properties of fluorophores. This may indeed be an advantage in terms of a robust performance of the decryption procedure. Still it is instrumentally demanding and less flexible in terms of mobility. I think that the undertaken revision and argumentation deserve a positive assessment and justify the acceptance of the manuscript.

Answer: Dear Reviewer #3, thank you very much for your positive feedback.